# Predictive validity of the quick Sequential Organ Failure Assessment (qSOFA) score for the mortality in patients with sepsis in Vietnamese intensive care units

Son Ngoc Do[1,2,3⦵], Chinh Quoc Luong[2,3,4⦵]*, My Ha Nguyen[5], Dung Thi Pham[6], Nga Thi Nguyen[7], Dai Quang Huynh[8,9], Quoc Trong Ai Hoang[10], Co Xuan Dao[1,2,3], Thang Dinh Vu[11], Ha Nhat Bui[12], Hung Tan Nguyen[13], Hai Bui Hoang[2,14], Thuy Thi Phuong Le[15], Lien Thi Bao Nguyen[16], Phuoc Thien Duong[17], Tuan Dang Nguyen[18], Vuong Hung Le[19], Giang Thi Tra Pham[20], Tam Van Bui[7], Giang Thi Huong Bui[1,2], Jason Phua[21,22], Andrew Li[22], Thao Thi Ngoc Pham[8,9], Chi Van Nguyen[2,4], Anh Dat Nguyen[2,4]

1 Center for Critical Care Medicine, Bach Mai Hospital, Hanoi, Vietnam, 2 Department of Emergency and Critical Care Medicine, Hanoi Medical University, Hanoi, Vietnam, 3 Faculty of Medicine, University of Medicine and Pharmacy, Vietnam National University, Hanoi, Vietnam, 4 Center for Emergency Medicine, Bach Mai Hospital, Hanoi, Vietnam, 5 Department of Health Organization and Management, Faculty of Public Health, Thai Binh University of Medicine and Pharmacy, Thai Binh, Vietnam, 6 Department of Nutrition and Food Safety, Faculty of Public Health, Thai Binh University of Medicine and Pharmacy, Thai Binh, Vietnam, 7 Department of Intensive Care and Poison Control, Vietnam-Czechoslovakia Friendship Hospital, Hai Phong, Vietnam, 8 Intensive Care Department, Cho Ray Hospital, Ho Chi Minh City, Vietnam, 9 Department of Critical Care, Emergency Medicine and Clinical Toxicology, Faculty of Medicine, Ho Chi Minh City University of Medicine and Pharmacy, Ho Chi Minh City, Vietnam, 10 Emergency Department, Hue Central General Hospital, Hue City, Thua Thien Hue, Vietnam, 11 Intensive Care Unit, People's Hospital 115, Ho Chi Minh City, Vietnam, 12 Intensive Care Unit, Bai Chay General Hospital, Quang Ninh, Vietnam, 13 Intensive Care Unit, Da Nang Hospital, Da Nang City, Vietnam, 14 Emergency and Critical Care Department, Hanoi Medical University Hospital, Hanoi Medical University, Hanoi, Vietnam, 15 Intensive Care Unit, Dong Da General Hospital, Hanoi, Vietnam, 16 Intensive Care Unit, Saint Paul General Hospital, Hanoi, Vietnam, 17 Intensive Care Unit, Can Tho Central General Hospital, Can Tho, Vietnam, 18 Intensive Care Unit, Vinmec Times City International Hospital, Hanoi, Vietnam, 19 Intensive Care Unit, Thai Nguyen National Hospital, Thai Nguyen, Vietnam, 20 Emergency Department, Thanh Nhan General Hospital, Hanoi, Vietnam, 21 FAST and Chronic Programmes, Alexandra Hospital, National University Health System, Singapore, Singapore, 22 Division of Respiratory and Critical Care Medicine, Department of Medicine, National University Health System, Singapore, Singapore

⦵ These authors contributed equally to this work.
* luongquocchinh@gmail.com

**Data Availability Statement:** All relevant data are within the paper and its Supporting Information files.

## Abstract

### Background

The simple scoring systems for predicting the outcome of sepsis in intensive care units (ICUs) are few, especially for limited-resource settings. Therefore, this study aimed to evaluate the accuracy of the quick Sequential (Sepsis-Related) Organ Failure Assessment (qSOFA) score in predicting the mortality of ICU patients with sepsis in Vietnam.

### Methods

We did a multicenter cross-sectional study of patients with sepsis (≥18 years old) presenting to 15 adult ICUs throughout Vietnam on the specified days (i.e., 9th January, 3rd April,

**Funding:** The author(s) received no specific funding for this work.

**Competing interests:** The authors have declared that no competing interests exist.

3rd July, and 9th October) representing the different seasons of 2019. The primary and secondary outcomes were the hospital and ICU all-cause mortalities, respectively. The area under the receiver operating characteristic curve (AUROC) was calculated to determine the discriminatory ability of the qSOFA score for deaths in the hospital and ICU. The cut-off value of the qSOFA scores was determined by the receiver operating characteristic curve analysis. Upon ICU admission, factors associated with the hospital and ICU mortalities were assessed in univariable and multivariable logistic models.

## Results

Of 252 patients, 40.1% died in the hospital, and 33.3% died in the ICU. The qSOFA score had a poor discriminatory ability for both the hospital (AUROC: 0.610 [95% CI: 0.538 to 0.681]; cut-off value: $\geq$2.5; sensitivity: 34.7%; specificity: 84.1%; $P_{AUROC}$ = 0.003) and ICU (AUROC: 0.619 [95% CI: 0.544 to 0.694]; cutoff value: $\geq$2.5; sensitivity: 36.9%; specificity: 83.3%; $P_{AUROC}$ = 0.002) mortalities. However, multivariable logistic regression analyses show that the qSOFA score of 3 was independently associated with the increased risk of deaths in both the hospital (adjusted odds ratio, AOR: 3.358; 95% confidence interval, CI: 1.756 to 6.422) and the ICU (AOR: 3.060; 95% CI: 1.651 to 5.671).

## Conclusion

In our study, despite having a poor discriminatory value, the qSOFA score seems worthwhile in predicting mortality in ICU patients with sepsis in limited-resource settings.

## Clinical trial registration

Clinical trials registry–India: CTRI/2019/01/016898

## Introduction

Sepsis is defined as life-threatening acute organ dysfunction caused by a dysregulated host response to infection [1]. It contributes to as much as 20.0% of all deaths worldwide [2], and mortality rates remain high at 30.0–45.0% [2–4]. In Asia, the overall hospital mortality of sepsis in the intensive care unit (ICU) fell from 44.5% (572/1285) to 36.6%; (1822/4980) over the past ten years [5, 6]; however, it is persistently high in lower-middle-income countries (LMICs), such as Indonesia (68.3%; 41/60) [7], Thailand (42.0%; 263/627) [8], and Vietnam (61.0%; 75/123) [9]. No reference standard exists that enables quick, precise diagnosis and prognosis of sepsis [1, 10]. In 2016, the Sepsis-3 Task Force suggested that an increase of 2 points in the Sequential (Sepsis-Related) Organ Failure Assessment (SOFA) score for patients with a suspected infection could be used as clinical criteria for sepsis [1]. This strategy was supported by content validity (SOFA reflects the facets of organ dysfunction) and predictive validity (the proposed criteria predict downstream events associated with the condition of interest) [11]. However, because many SOFA variables are not routinely measured or are not available, the utility of SOFA is constrained both inside and outside the ICU in settings with limited resources.

The Sepsis-3 Task Force also noted that the quick SOFA (qSOFA) score, a combination of respiratory rate, mental state, and systolic blood pressure, had strong predictive validity for

sepsis in patients outside the ICU [11]. Since then, it has been prospectively investigated in a number of settings, including the emergency department (ED), normal ward, and ICU. However, the results are conflicting as to whether it can reliably identify these populations or predict their risk of dying [12–16]. Because the qSOFA score only requires a clinical examination, it may be especially useful in settings with limited resources. The applicability of this score to various infection types, hospital departments (ED, ward, ICU), and nations is still up for debate.

With 96.462 million inhabitants, Vietnam is an LMIC, 15th in the world and third in Southeast Asia in terms of population [17]. Southeast Asia's hotspot for newly emerging infectious diseases, such as SARS-CoV [18], A(H5N1) avian influenza [19, 20], and ongoing COVID-19 outbreaks worldwide [21], is Vietnam. Other significant causes of sepsis in ICUs across Vietnam include severe dengue [22], *Streptococcus suis* infection [23], malaria [24], and increased antibiotic resistance [25, 26]. Vietnam continues to struggle to provide sufficient resources or adequate diagnostic, prognostic, and treatment approaches for patients with sepsis [27, 28], despite its recent economic growth spurt [29]. Additionally, central hospitals in Vietnam's healthcare system are in charge of receiving patients who need assistance receiving care in local hospital settings [30]. Consequently, the diagnosis, prognosis, and initiation of treatment for patients with sepsis are often delayed.

In resource-limited settings, the early identification of infected patients who may go on to develop sepsis or may be at risk of death from sepsis using simple scoring systems as a way to make a better decision on the treatment of these patients. The aim of this study, therefore, was to evaluate the accuracy of qSOFA score in predicting the mortality of ICU patients with sepsis in Vietnam.

## Methods

### Study design and setting

This multicenter observational, cross-sectional, point prevalence study is part of the Management of Severe sepsis in Asia's Intensive Care unitS (MOSAICS) II study [6, 31, 32], which enrolled patients on 9th January (Winter), 3rd April (Spring), 3rd July (Summer), and 9th October (Autumn) of 2019. In this study, we used only data from Vietnam. A total of 15 adult ICUs (excluding predominantly neurosurgical, coronary, and cardiothoracic ICUs) participating in the MOSAICS II study from 14 hospitals, of which 5 are central and 9 are provincial, district, or private hospitals, throughout Vietnam. Each ICU had one or two representatives who were part of the local study team and the MOSAICS II study group, as shown in eAppendix 2 of a previously published paper [6]. Participation was voluntary and unfunded.

### Participants

All patients admitted to participating ICUs on one of the four days (i.e., January 9th, April 3rd, July 3rd, and October 9th, 2019) which represented the different seasons of 2019 were screened for eligibility. We included all patients, aged ≥18 years old, who were admitted to the ICUs for sepsis, and who were still in the ICUs from 00:00 hour to 23:59 hour of the study days. We defined sepsis as infection with a Sequential Organ Failure Assessment (SOFA) score ≥2 from baseline (assumed to be 0 for patients without prior organ dysfunction) [1].

### Data collection

We used a standardized classification and case record form (CRF) to collect data on common variables as shown in S1 File. The data dictionary of the MOSAICS II study is available as an online supplement of previously published papers [6, 32]. Data was entered by the representatives

of the participating hospitals into the database of the MOSAICS II study via the password-protected online CRFs. We checked the data for implausible outliers and missing fields and contacted ICU representatives for clarification. We then merged the data sets for the 14 hospitals.

## Variables

We included variables based on the CRF which is available as shown in S1 File, such as information on: participating hospitals (e.g., central hospital), demographics (e.g., sex, age), comorbidities (e.g., cardiovascular disease, chronic lung disease, chronic neurological disease, chronic kidney disease, peptic ulcer disease, chronic liver disease, diabetes mellitus, and solid malignant tumors), sites of infection (e.g., respiratory, urinary tract, abdominal, neurological, and skin or cutaneous sites), and vital signs (e.g., Glasgow coma score [GCS], heart rate [HR], body temperature, respiratory rate [RR], and blood pressure [BP]) and severity of illness upon ICU admission (e.g., SOFA score, qSOFA score, and septic shock). The qSOFA score ranges from 0 to 3, with one point allocated for each of the following clinical signs: systolic BP ≤100 mmHg, RR ≥22 breaths/min, and altered mental status from baseline (assumed to be normal for patients with a GCS of 15) [11]. Septic shock was defined as a clinical construct of sepsis with persisting hypotension requiring vasopressors to maintain mean arterial pressure ≥65 mmHg and having a serum lactate level >2 mmol/L (18mg/dL) despite adequate volume resuscitation [1]. We also collected information on life-sustaining treatments provided during the ICU stay, e.g., mechanical ventilation (MV) and renal replacement therapy (RRT). We followed all patients till hospital discharge, death in the ICU/hospital, or up to 90 day post-enrollment, whichever was earliest.

## Outcomes

The primary outcome was hospital all-cause mortality (hospital mortality). We also examined the following secondary outcomes: ICU all-cause mortality (ICU mortality), and ICU and hospital lengths of stay (LOS).

## Sample size

In this cross-sectional study, the primary outcome was hospital mortality. Therefore, based on the hospital mortality rate (61.0%) of our cohort reported in a previously published study [9], we used the formula to find the minimum sample size for estimating a population proportion, with a confidence level of 95% and a confidence interval (margin of error) of ±6.03%, and an assumed population proportion of 61.0%. As a result, our sample size should be at least 252 patients. Therefore, our sample size was large enough which reflects a normal distribution.

$$n = \frac{z^2 x \, \hat{p}(1-\hat{p})}{\varepsilon^2}$$

where:

 *z is the z score (z score for a 95% confidence level is 1.96)*
 *$\varepsilon$ is the margin of error ($\varepsilon$ for a confidence interval of ± 6.03% is 0.0603)*
 *$\hat{p}$ is the population proportion ($\hat{p}$ for a population proportion of 61.0% is 0.61*
 *n is the sample size*

## Statistical analyses

We used IBM® SPSS® Statistics 22.0 (IBM Corp., Armonk, United States of America) for data analysis. We report data as numbers and percentages for categorical variables and medians and quartiles (Q1-Q3) in the case of non-normal distribution or means and standard

deviations (SDs) in the case of normal distribution for continuous variables. Comparisons were made between survival and death in the hospital for each variable, using the Chi-squared test or Fisher exact test for categorical variables and the Mann-Whitney U test, Kruskal-Wallis test, one-way analysis of variance for continuous variables.

Receiver operator characteristic (ROC) curves were plotted and the areas under the receiver operating characteristic curve (AUROC) were calculated to determine the discriminatory ability of the qSOFA score for deaths in the hospital and ICU. Additionally, to evaluate the predictive validity of the qSOFA-65, defined as including the age criterion ≥65 years to the qSOFA score [33], for the mortality in ICU patients with sepsis, we also used the ROC curves and AUROCs to determine the discriminatory ability of the qSOFA-65 score for deaths in the hospital and ICU. The cut-off value of the qSOFA score was determined by the ROC curve analysis and defined as the cut-off point with the maximum value of Youden's index (i.e., sensitivity + specificity—1). Based on the cut-off value of the qSOFA score, the patients were classified into two groups. Additionally, based on the cut-off value (≥ 2.0) of the originally-suggested qSOFA score [1, 11], patients were assigned to two groups: either the qSOFA scores of 0 to 1 or the qSOFA scores of 2 to 3.

Upon ICU admission, we assessed factors associated with death in the hospital using logistic regression analysis. To reduce the number of predictors and the multicollinearity issue and resolve the overfitting, we used different ways to select variables as follows: (a) we put all variables of participating hospitals, demographics, and baseline characteristics into the univariable logistic regression model; (b) we selected variables if the P-value was <0.25 in the univariable logistic regression analysis between survival and death in the hospital, as well as those that are clinically crucial (e.g., sex, age), to put in the multivariable logistic regression model. These variables included participating hospitals (i.e., central hospitals), demographics (i.e., sex, age), documented comorbidities (i.e., cardiovascular disease, chronic neurological disease, solid malignant tumors), sites of infection (i.e., urinary tract, skin, or cutaneous sites), and severity of illness (i.e., qSOFA score). Using a stepwise backward elimination method, we started with the full multivariable logistic regression model that included the selected variables. This method then deleted the variables stepwise from the full model until all remaining variables were independently associated with the risk of death in the hospital in the final model. Similarly, we used these methods of variable selection and analysis for assessing factors associated with death in the ICU. We presented the adjusted odds ratios (AORs) and 95% confidence intervals (CIs) in the multivariable logistic regression model.

For all analyses, significance levels were two-tailed, and we considered P < 0.05 as statistically significant. No adjustments for multiple testing were required in this study.

### Ethical issues

The Bach Mai Hospital Scientific and Ethics Committees approved this study (approval number: 2919/QD–BM; project code: BM-2017-883-89). We also obtained permission from the heads of institutions and departments of all participating hospitals and their respective institutional review boards wherever available. The study was conducted according to the principles of the Declaration of Helsinki. The Bach Mai Hospital Scientific and Ethics Committees waived written informed consent for this noninterventional study, and public notification of the study was made by public posting. The authors who did the data analysis kept the data sets in password-protected systems and we present anonymized data.

### Results

Data on 252 patients with sepsis were submitted to the database of the MOSAICS II study (S1 Fig as shown in S2 File), in which there were little missing data. Of these patients, 64.3% (162/

252) were men and the median age was 65 years (Q1-Q3: 52–77) (Table 1), of which the median qSOFA score was 2 (Q1-Q3: 1–2) and the median SOFA score was 7 (Q1-Q3: 5–10) at the time of ICU admission. Table 1 also shows that the most common documented comorbidities included cardiovascular disease (31.0%; 78/252), diabetes mellitus (26.6%; 67/252), and chronic neurological disease (14.3%; 36/252) and the most common sites of infection included respiratory (56.7%; 143/252), abdominal cavity (24.2%; 61/252), urinary tract (14.7%; 37/252) and skin or cutaneous sites (7.5%; 19/252). Gram-negative bacteria were isolated in 61.9% (156/252) of patients, with *Acinetobacter baumannii* (17.9%; 45/252) predominating (S1 and S3 Tables as shown in S2 File). Table 2 shows that MV was provided for 68.9% (173/251) of patients and RRT for 40.2% (101/251). Overall, 40.1% (101/252) of patients with sepsis died during the hospital stay, 33.3% (84/252) of whom died in the ICU (Tables 1 and 2). The median ICU and hospital LOS were 10 (Q1-Q3: 6–18) and 16 (Q1-Q3: 10–25) days, respectively (Table 2). The baseline characteristics and life-sustaining treatments during ICU stay were compared between patients who survived and patients who died in the hospital and ICU, as shown in Tables 1 and 2, S1-S4 Tables in S2 File).

In this study, the qSOFA score had a poor discriminatory ability for both the hospital (Fig 1; AUROC: 0.610 [95% CI: 0.538–0.681]; cut-off value: ≥2.5; sensitivity: 34.7%; specificity: 84.1%; $P_{AUROC}$ = 0.003) and ICU (S2 Fig as shown in S2 File; AUROC: 0.619 [95% CI: 0.544–0.694]; cut-off value: ≥2.5; sensitivity: 36.9%; specificity: 83.3%; $P_{AUROC}$ = 0.002) mortalities. We also have added the age criterion ≥65 years to the qSOFA score and compared this extended score (qSOFA-65) to the original score (qSOFA). However, like the qSOFA score (S3 and S4 Figs as shown in S2 File), the qSOFA-65 also had a poor discriminatory ability for both the hospital (S3 Fig as shown in S2 File; AUROC: 0.548 [95% CI: 0.476–0.619]; cut-off value: ≥2.5; sensitivity: 53.5%; specificity: 53.0%; $P_{AUROC}$ = 0.201) and ICU (S4 Fig as shown in S2 File; AUROC: 0.562 [95% CI: 0.488–0.636]; cut-off value: ≥2.5; sensitivity: 56.0%; specificity: 53.6%; $P_{AUROC}$ = 0.107) mortalities.

Based on the cut-off value (≥2.5) of the qSOFA score (Fig 1, S2 Fig as shown in S2 File), the patients were classified into two groups, of which the qSOFA score of 3 reflects patients who met all three criteria of the qSOFA score. In the univariable logistic regression analyses, the qSOFA score of 3 was significantly associated with the increased risk of deaths in both the hospital (OR: 2.806; 95% CI: 1.542–5.106) and ICU (OR: 2.925; 95% CI: 1.604–5.333) (Table 3, S5 and S6 Tables as shown in S2 File). Furthermore, the multivariable logistic regression analyses show that the qSOFA score of 3 was independently associated with the increased risk of deaths in both the hospital (AOR: 3.358; 95% CI: 1.756 to 6.422) and ICU (AOR: 3.060; 95% CI: 1.651 to 5.671) (Table 3, S7 and S8 Tables as shown in S2 File). When we replaced the qSOFA score of 3 with the qSOFA score of 2 to 3 in these multivariable logistic regression analyses, we found that the effect size for the qSOFA score of 2 to 3, although, was more modest than those for the qSOFA score of 3, the qSOFA score of 2 to 3 was still an independent predictor of deaths in both the hospital (AOR: 2.101; 95% CI: 1.118 to 3.951) and ICU (AOR: 2.222; 95% CI: 1.153 to 4.282) (S9 and S10 Tables as shown in S2 File).

## Discussion

The present study revealed that about a third (33.3%; 84/252) of patients with sepsis died in the ICU, and two-fifths (40.1%; 101/252) died in the hospital. The qSOFA score had a poor discriminatory ability for hospital and ICU mortality. However, a qSOFA score of 3 was independently associated with the increased risk of death in both hospitals and ICUs.

Our figure for the hospital mortality rate was lower than the figures reported previously from LMICs in Southeast Asia, such as Indonesia (68.3%; 41/60) [7], Thailand (42%; 263/627)

**Table 1. Demographics and baseline characteristics of intensive care unit patients with sepsis according to hospital survivability.**

| Variables | All cases | Survived | Died | P-value[a] |
|---|---|---|---|---|
| | n = 252 | n = 151 | n = 101 | |
| **Participating hospital, no. (%)** | | | | 0.041 |
| Central hospitals | 113 (45.6) | 61 (40.4) | 54 (53.5) | |
| Provincial, district, or private hospitals | 137 (54.5) | 90 (59.6) | 47 (46.5) | |
| **Demographics** | | | | |
| Age (year), median (Q1-Q3) | 65 (52–77) | 65 (53–76) | 65 (52–78) | 0.810** |
| Age (year), no. (%) | | | | 0.939 |
| < 65 | 123 (48.8) | 74 (49.0) | 49 (48.5) | |
| ≥ 65 | 129 (51.2) | 77 (51.0) | 52 (51.5) | |
| Sex (male), no. (%) | 162 (64.3) | 93 (61.6) | 69 (68.3) | 0.275 |
| **Documented comorbidities** | | | | |
| Cardiovascular disease, no. (%) | 78 (31.0) | 41 (27.2) | 37 (36.6) | 0.111 |
| Chronic lung disease, no. (%) | 30 (11.9) | 18 (11.9) | 12 (1.9) | 0.992 |
| Chronic neurological disease, no. (%) | 36 (14.3) | 28 (18.5) | 8 (7.9) | 0.018 |
| Chronic kidney disease, no. (%) | 23 (9.1) | 14 (9.3) | 9 (8.9) | 0.922 |
| Peptic ulcer disease, no. (%) | 9 (3.6) | 5 (3.3) | 4 (4.0) | >0.999* |
| Chronic liver disease, no. (%) | 27 (10.7) | 14 (9.3) | 13 (12.9) | 0.365 |
| Diabetes mellitus, no. (%) | 67 (26.6) | 40 (26.5) | 27 (26.7) | 0.966 |
| HIV infection, no. (%) | 0 (0.0) | 0 (0.0) | 0 (0.0) | NA |
| Connective tissue disease, no. (%) | 3 (1.2) | 2 (1.3) | 1 (1.0) | >0.999* |
| Immunosuppression, no. (%) | 10 (4.0) | 7 (4.6) | 3 (3.0) | 0.744 |
| Hematological malignancies, no. (%) | 5 (2.0) | 3 (2.0) | 2 (2.0) | >0.999* |
| Solid malignant tumours, no. (%) | 12 (4.8) | 6 (4.0) | 6 (5.9) | 0.551* |
| **Vital signs** | | | | |
| GCS, median (Q1-Q3) | 13 (9–15) | 14 (10–15) | 10 (8–14) | <0.001** |
| HR (beats per min), median (Q1-Q3) | 110 (95–126) | 110 (92–125) | 110 (100–130) | 0.083** |
| Temperature (˚C), mean (SD) | 37.79 (1.01) | 37.80 (1.08) | 37.77 (0.91) | 0.871** |
| MBP (mmHg), mean(SD) | 75.82 (22.08) | 79.75 (22.88) | 69.93 (19.51) | 0.002** |
| SBP (mmHg), mean (SD) | 106.45 (29.96) | 111.39 (29.44) | 99.07 (29.35) | 0.004** |
| RR (breaths per min), median (Q1-Q3) | 25 (22–30) | 25 (22–30) | 25 (20–30) | 0.693** |
| **Blood investigations** | | | | |
| Total WBC (x10$^9$/L), mean (SD) | 15.73 (9.20) | 15.63 (8.67) | 15.88 (9.98) | 0.914** |
| PLT (x10$^9$/L), mean (SD) | 185.98 (137.85) | 200.71 (129.67) | 163.95 (147.15) | 0.002** |
| Hb (g/dL), mean (SD) | 11.14 (2.59) | 11.36 (2.68) | 10.82 (2.44) | 0.088** |
| Hct (%), mean (SD) | 34.31 (7.75) | 35.08 (7.92) | 33.17 (7.38) | 0.031** |
| $K^+$ (mmol/L), mean (SD) | 3.89 (0.79) | 3.90 (0.80) | 3.87 (0.77) | 0.865** |
| $Na^+$ (mmol/L), mean (SD) | 136.05 (8.24) | 135.62 (8.81) | 136.69 (7.80) | 0.068** |
| Creatinine (μmol/L), mean (SD) | 187.85 (151.92) | 186.15 (171.60) | 190.38 (117.27) | 0.030** |
| Bilirubin (μmol/l), mean (SD) | 32.80 (61.49) | 31.74 (72.67) | 34.35 (40.09) | 0.007** |
| pH, mean (SD) | 7.37 (0.50) | 7.41 (0.64) | 7.32 (0.14) | 0.004** |
| $PaO_2$ (mmHg), mean (SD) | 116.17 (74.28) | 110.23 (56.25) | 124.73 (94.07) | 0.665** |
| $PaO_2/FiO_2$ ratio, mean (SD) | 262.48 (149.58) | 281.52 (149.39) | 235.26 (146.32) | 0.003** |
| **Severity of illness scores** | | | | |
| qSOFA score, median (Q1-Q3) | 2 (1–2) | 2 (1–2) | 2 (2–3) | 0.001** |
| qSOFA score, no. (%) | | | | 0.006 |
| 0 | 9 (3.6) | 6 (4.0) | 3 (3.0) | |
| 1 | 60 (23.8) | 42 (27.8) | 18 (17.8) | |

*(Continued)*

**Table 1.** (Continued)

| Variables | All cases | Survived | Died | P-value[a] |
|---|---|---|---|---|
| | n = 252 | n = 151 | n = 101 | |
| 2 | 124 (49.2) | 79 (52.3) | 45 (44.6) | |
| 3 | 59 (23.4) | 24 (15.9) | 35 (34.7) | |
| SOFA score, median (Q1-Q3) | 7 (5–10) | 6 (4–9) | 9 (6–12) | <0.001** |
| Septic Shock, no. (%) | 74 (29.4) | 35 (23.2) | 39 (38.6) | 0.008 |
| **Site of Infection** | | | | |
| Respiratory, no. (%) | 143 (56.7) | 82 (54.3) | 61 (60.4) | 0.339 |
| Urinary tract, no. (%) | 37 (14.7) | 30 (19.9) | 7 (6.9) | 0.004 |
| Abdominal, no. (%) | 61 (24.2) | 34 (22.5) | 27 (26.7) | 0.444 |
| Neurological, no. (%) | 12 (4.8) | 8 (5.3) | 4 (4.0) | 0.767* |
| Bones or joints, no. (%) | 2 (0.8) | 2 (1.3) | 0 (0.0) | 0.518* |
| Skin or cutaneous sites, no. (%) | 19 (7.5) | 7 (4.6) | 12 (11.9) | 0.033 |
| Intravascular catheter, no. (%) | 1 (0.4) | 1 (0.7) | 0 (0.0) | >0.999* |
| Infective endocarditis, no. (%) | 1 (0.4) | 0 (0.0) | 1 (1.0) | 0.401* |
| Primary bacteremia, no. (%) | 7 (2.8) | 5 (3.3) | 2 (2.0) | 0.705* |
| Systemic, no. (%) | 6 (2.4) | 4 (2.6) | 2 (2.0) | >0.999* |

[a]Comparison between the patients who survived and died using Chi-squared test

*Fisher's exact test

**Mann–Whitney U test.

Abbreviations

**FiO₂**: fraction of inspired oxygen; **GCS**: Glasgow coma scale; **Hb**: hemoglobin; **Hct**: hematocrit; **HIV**: human immunodeficiency viruses; **HR**: heart rate; **MBP**: mean arterial blood pressure; **n**, total number of patient; **NA**, not available; **no.**, total number of patients recorded if a variable was given; **PaO₂**: partial pressure of arterial oxygen; **PLT**: platelet; **Q**: quartile; **qSOFA**: quick Sequential (Sepsis-Related) Organ Failure Assessment; **RR**: respiratory rate; **SBP**: systolic blood pressure; **SD**: standard deviation; **SOFA**: Sequential (Sepsis-Related) Organ Failure Assessment; **WBC**: white blood cell.

See (S1 Table as shown in S2 File) for additional information.

[8], and Vietnam (61.0%; 75/123) [9], and lower than the figure reported in the MOSAICS I study (44.5%; 572/1285) [5]. These findings might be due to the definition and management of sepsis having evolved tremendously to improve survival in patients with sepsis and septic shock in the past decade [1, 6, 10, 34–36]. However, our study showed that the ICU and hospital mortality rates were higher than rates reported in the international Extended Study on Prevalence of Infection in Intensive Care (EPIC III) study (28% [99/352] and 31.1% [110/352] in LMICs, 26.4% [821/3114] and 32.7% [1019/3114] in upper-middle-income countries [UMICs], and 21.3% [950/4470] and 28.5% [1275/4470] in high-income countries [HICs]) [37]. These differences might be because the EPIC III study included ICU-acquired infection and not specifically sepsis [37]. Despite the distinct inclusion criteria, our median SOFA score upon admission into the ICU was consistent with those reported in the EPIC III study (7 points [Q1-Q3: 4–11] in LMICs/UMICs/HICs) [37]. However, invasive organ support therapies during ICU stay (i.e., MV and RRT) were more often given to patients in our study than that to patients in the EPIC III study (54.4% [4377/8045] and 15.7% [1253/8045]) [37]. Previous studies showed that MV at any time during the ICU stay was a crucial predictor of death [32, 38]. Additionally, the utilization of RRT at any time during the ICU stay was also associated with increased mortality [32, 38–41]. Furthermore, one of the most dangerous pathogens was *Acinetobacter baumannii*, which was much more commonly isolated from patients in the present study than those in HICs (4.4%; 137/3113) of the EPIC III study [37]. The previous studies showed that *Acinetobacter baumannii* infection was often due to a lack of strict

**Table 2. Life-sustaining treatments during intensive care unit (ICU) stay and outcomes of ICU patients with sepsis according to hospital survivability.**

| Variables | All cases | Survived | Died | P-value[a] |
|---|---|---|---|---|
| | n = 252 | n = 151 | n = 101 | |
| **Life-sustaining treatments during ICU stay** | | | | |
| Respiratory support, no. (%) | | | | |
| Mechanical ventilation[b] | 173 (68.9) | 82 (54.7) | 91 (90.1) | <0.001 |
| Non-invasive ventilation[b] | 20 (8.0) | 13 (8.7) | 7 (6.9) | 0.618 |
| High-flow nasal oxygen[b] | 38 (15.1) | 29 (19.3) | 9 (8.9) | 0.024 |
| Additional ICU support, no. (%) | | | | |
| Vasopressors/inotropes | 163 (64.7) | 82 (54.3) | 81 (80.2) | <0.001 |
| Renal replacement therapy[b] | 101 (40.2) | 43 (28.7) | 58 (57.4) | <0.001 |
| Red blood cell transfusion[b] | 93 (37.1) | 48 (32.0) | 45 (44.6) | 0.043 |
| Platelet transfusion[b] | 50 (19.9) | 20 (13.3) | 30 (29.7) | 0.001 |
| Fresh frozen plasma transfusion[b] | 58 (23.1) | 28 (18.7) | 30 (29.7) | 0.042 |
| Surgical source control[b] | 25 (10.0) | 19 (12.7) | 6 (5.9) | 0.081 |
| Non-surgical source control[b] | 78 (31.1) | 54 (36.0) | 24 (23.8) | 0.040 |
| **Outcomes** | | | | |
| Patient status, no. (%) | | | | <0.001* |
| Alive upon current hospital discharge, no. (%) | 150 (59.5) | 150 (99.3) | 0 (0.0) | |
| Alive upon discharge from current ICU stay, but died in current hospital stay, no. (%) | 17 (6.7) | 0 (0.0) | 17 (16.8) | |
| Alive upon discharge from current ICU stay, but still in current hospital stay after 90 days, no. (%) | 1 (0.4) | 1 (0.7) | 0 (0.0) | |
| Still in current ICU stay after 90 days, no. (%) | 0 (0.0) | 0 (0.0) | 0 (0.0) | |
| Died in current ICU stay, no. (%) | 84 (33.3) | 0 (0.0) | 84 (83.2) | |
| Length of stay, median days (Q1-Q3) | | | | |
| Hospital | 16 (10–25) | 17 (11–24.25) | 13 (7–26) | 0.027** |
| ICU | 10 (6–18) | 10.5 (6–17) | 10 (5–21) | 0.740** |

[a]Comparison between the patients who survived and died using Chi-squared test

*Fisher's exact test

**Mann–Whitney U test.

[b]Missing details from one patient.

Abbreviations

**ICU**: intensive care unit; **n**, total number of patient; **no.**, total number of patients recorded if a variable was given; **Q**: quartile.

See (S2 Table as shown in S2 File) for additional information.

infection control bundles [42] and associated with an increased risk of death [43, 44]. All these findings indicated that the patients, pathogens, and clinical capacity to manage sepsis differ considerably among regions, especially between HIC and LMIC settings, and might explain that our proportions for the ICU and hospital mortality were higher than the rates reported in EPIC III study [37].

In this study, we found a poor ability of the qSOFA score to predict the hospital and ICU mortalities. Despite the first strong predictive validity of the qSOFA score for hospital mortality in patients with suspected infection outside of the ICU [11], there are conflicting data regarding its ability to accurately predict the risk of deaths from sepsis in different populations [12–16]. In an international prospective cohort study of patients presenting to the ED with suspected infection, the predictive validity of qSOFA score for the hospital mortality was acceptable, with an AUROC of 0.800 (95% CI, 0.740 to 0.850), as well as is similar to that of the SOFA score with an AUROC of 0.770 (95% CI, 0.710 to 0.820) [12]. In contrast, another retrospective cohort study shows a poor ability of the qSOFA score for predicting 28-day mortality

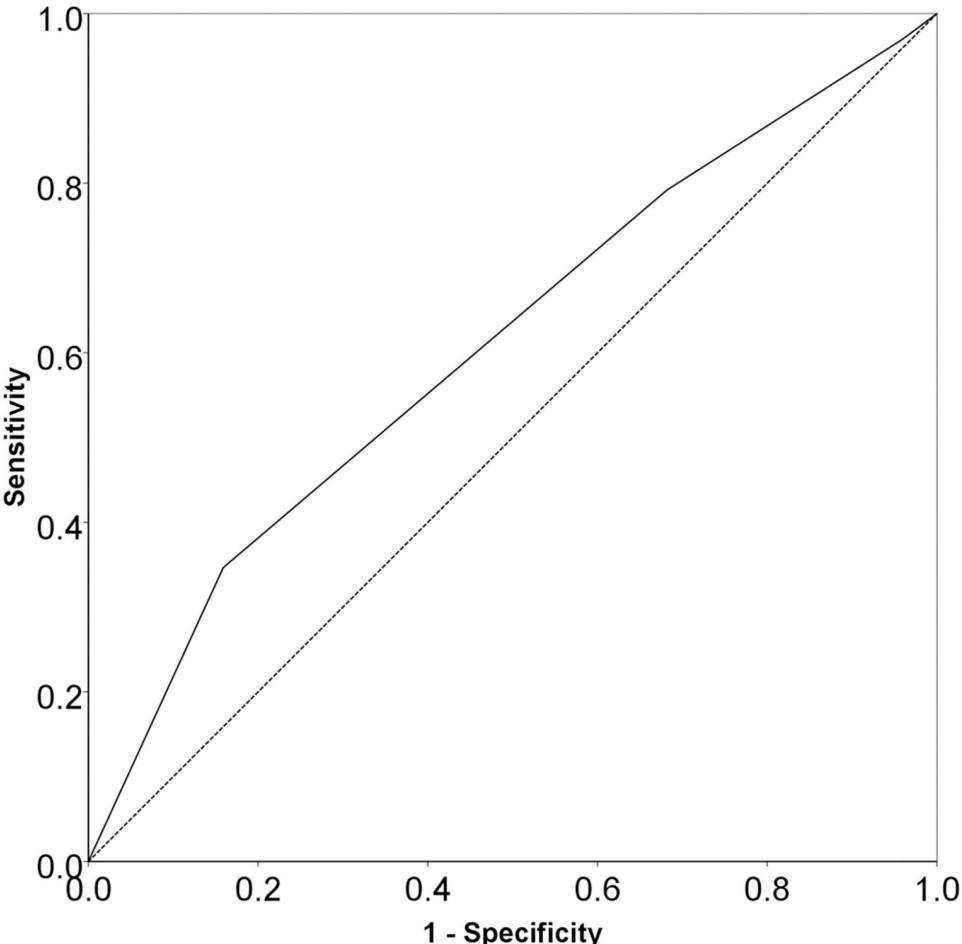

**Fig 1. The area under the ROC curves of the qSOFA score (AUROC: 0.610 [95% CI: 0.538–0.681]; cut-off value: ≥2.5; sensitivity: 34.7%; specificity: 84.1%; P$_{AUROC}$ = 0.003) for predicting the hospital mortality in ICU patients with sepsis in Vietnamese ICUs.** Abbreviations: **AUROC**, area under the receiver operating characteristic curve; **CI**, confidence interval; **ICU**, intensive care unit; **qSOFA**, Quick Sequential Organ Failure Assessment; **ROC**, receiver operating characteristic; **SOFA**, Quick Sequential Organ Failure Assessment.

in critically ill patients with sepsis on ED arrival, with AUROC of 0.580 (95% CI 0.550 to 0.620) [14]. A large retrospective cohort study also shows the poor discrimination for hospital mortality of the qSOFA score with an AUROC of 0.607 [99% CI, 0.603 to 0.611]) in ICU patients with sepsis [16]. Despite the poor predictive validity of the qSOFA score for the hospital and ICU mortalities among ICU patients with sepsis, our study shows that the predictive validity for hospital mortality of qSOFA score above baseline risk was similar to those of the originally-suggested qSOFA score (AUROC = 0.660; 95% CI, 0.640–0.680; cut-off value: ≥2.0) in patients with sepsis in the ICU [1, 11]. Moreover, a recent retrospective population-based cohort study [33] also shows that the poor predictive validity of the qSOFA-65 (AUROC = 0.690; 95% CI, 0.690–0.690) for hospital mortality in patients with community-acquired pneumonia was better than those of the qSOFA-65 for hospital mortality in our ICU patients with sepsis. This difference might be due to a different population between the two studies. The present study shows that the qSOFA score of 3 upon ICU admission was independently associated with the increased risk of death in both the hospital and ICU. Moreover, although the effect size for the qSOFA score of 2 to 3 was more modest than those for the

**Table 3. Factors associated with mortality in patients with sepsis upon Vietnamese intensive care unit admission, 2019: Logistic regression analyses.**

| Factors | Univariable logistic regression analyses[a] | | | | Multivariable logistic regression analyses[b] | | | |
|---|---|---|---|---|---|---|---|---|
| | OR | 95% CI for OR | | P-value | AOR | 95% CI for AOR | | P-value |
| | | Lower | Upper | | | Lower | Upper | |
| **Factors associated with hospital mortality in patients with sepsis** | | | | | | | | |
| *Participating hospital* | | | | | | | | |
| Center hospitals[c] | 1.695 | 1.020 | 2.819 | 0.042 | 1.992 | 1.110 | 3.573 | 0.021 |
| *Demographics* | | | | | | | | |
| Age of 65 years or older | 1.020 | 0.616 | 1.688 | 0.939 | NA | NA | NA | NA |
| Sex (male) | 1.345 | 0.790 | 2.290 | 0.275 | NA | NA | NA | NA |
| *Documented comorbidities* | | | | | | | | |
| Cardiovascular disease | 1.551 | 0.903 | 2.664 | 0.112 | 2.011 | 1.086 | 3.721 | 0.026 |
| Chronic neurological disease | 0.378 | 0.165 | 0.867 | 0.022 | 0.387 | 0.152 | 0.983 | 0.046 |
| Solid malignant tumors | 1.526 | 0.478 | 4.873 | 0.475 | NA | NA | NA | NA |
| *Site of Infection* | | | | | | | | |
| Urinary Tract | 0.300 | 0.126 | 0.714 | 0.006 | 0.270 | 0.106 | 0.686 | 0.006 |
| Skin or Cutaneous Sites | 2.774 | 1.053 | 7.309 | 0.039 | 3.013 | 1.059 | 8.571 | 0.039 |
| *Severity of illness score* | | | | | | | | |
| qSOFA score of 3 | 2.806 | 1.542 | 5.106 | 0.001 | 3.358 | 1.756 | 6.422 | <0.001 |
| Constant | | | | | 0.350 | | | <0.001 |
| **Factors associated with intensive care unit mortality in patients with sepsis** | | | | | | | | |
| *Participating hospital* | | | | | | | | |
| Central hospitals[c] | 1.211 | 0.716 | 2.048 | 0.475 | NA | NA | NA | NA |
| *Demographics* | | | | | | | | |
| Age of 65 years or older | 1.000 | 0.592 | 1.689 | >0.999 | NA | NA | NA | NA |
| Sex (male) | 0.728 | 0.417 | 1.272 | 0.265 | NA | NA | NA | NA |
| *Documented comorbidities* | | | | | | | | |
| Cardiovascular disease | 1.506 | 0.863 | 2.627 | 0.150 | NA | NA | NA | NA |
| Chronic neurological disease | 0.526 | 0.229 | 1.212 | 0.131 | NA | NA | NA | |
| Solid malignant tumors | 2.077 | 0.649 | 6.648 | 0.218 | NA | NA | NA | NA |
| *Site of Infection* | | | | | | | | |
| Urinary Tract | 0.340 | 0.136 | 0.851 | 0.021 | 0.318 | 0.123 | 0.822 | 0.018 |
| Skin or cutaneous sites | 2.387 | 0.931 | 6.123 | 0.070 | 2.365 | 0.893 | 6.264 | 0.083 |
| *Severity of illness score* | | | | | | | | |
| qSOFA score of 3 | 2.925 | 1.604 | 5.333 | <0.001 | 3.060 | 1.651 | 5.671 | <0.001 |
| Constant | | | | | 0.400 | | | <0.001 |

[a]Each variable of the hospital and baseline characteristics was analyzed in the univariable logistic regression model and was selected for the multivariate logistic regression model if the P-value was <0.25 between survival and death, as well as those that are clinically crucial.

[b]All selected variables were included in the multivariable logistic regression model with the stepwise backward elimination method. Variables, then, were deleted one by one from the full model until all remaining variables were independently associated with the risk of death in the final model.

[c]Center hospitals included the Thai Nguyen, Bach Mai, Hue, Cho Ray, and Can Tho hospitals. Abbreviations: **AOR**: adjusted odds ratio; **CI**: confidence interval; **NA**, not available; **OR**: odds ratio; **qSOFA**: quick Sequential (Sepsis-Related) Organ Failure Assessment.

qSOFA score of 3 in our study, the qSOFA score of 2 to 3 was still independently associated with the increased risk of death in both the hospital and ICU. Our findings are consistent with the results of a secondary analysis of 9 data sets from 8 cohort studies in LMICs, of which higher qSOFA scores were associated with a higher risk of death, but predictive validity varied significantly among cohorts which limited the interpretation of the results [45]. Therefore, our

study suggests that despite having a poor discriminatory value, the qSOFA score seems worthwhile in predicting mortality in ICU patients with sepsis in limited-resource settings.

## Strengths and limitations

An advantage of the present study was data from the multicenter, which had little missing data (S11 Table as shown in S2 File). However, the present study has some limitations as follows: (i) *Firstly*, due to the absence of a national registry of ICUs to allow systematic recruitment of units, we used a snowball method to identify suitable units, which might have led to the selection of centers with a greater interest in sepsis management. Therefore, our data are subject to selection bias [46] and might not reflect intensive care throughout Vietnam; (ii) *Secondly*, due to the study's real-world nature, we did not sufficiently protocolize microbiological investigations. Moreover, we mainly evaluated resources utilized in ICUs; therefore, the data detailing the life-sustaining treatments (e.g., fluid balance, administration of steroids, and modalities of RRT and MV) were unavailable; (iii) *Thirdly*, to improve the feasibility of performing the study in busy ICUs, we opted not to collect data on antibiotic resistance and appropriateness; (iv) *Fourthly*, due to our independent variables (e.g., qSOFA) that might be associated with primary outcome (hospital mortality) only measured upon ICU admission, the mixed-effects logistic regression model could not be used to predict discrete outcome variables measured at two different times, i.e., inside and outside the ICU setting. (v) *Finally*, although the sample size was large enough, the confidence interval was slightly wide (±6.03%), which might influence the normal distribution of the sample. Thus, further studies with larger sample sizes might be needed to consolidate the conclusions.

In conclusion, this was a selected cohort of patients with sepsis admitted to the ICUs in Vietnam with high mortality. The qSOFA score had a poor discriminatory ability for mortality; however, the effect size for the qSOFA score of 2 to 3, although, was more modest than those for the qSOFA score of 3, both these score groups were independently associated with the increased risk of deaths in both the hospital and the ICU. Based on the results of our study, despite having a poor discriminatory value, the qSOFA score seems worthwhile in predicting mortality in ICU patients with sepsis in limited-resource settings. However, further studies of the same type are needed by focusing more on the newer simple scoring systems to improve the predictive validity for the outcomes of patients with sepsis.

## Supporting information

**S1 File.**
(PDF)

**S2 File.**
(PDF)

**S1 Dataset.**
(XLSX)

## Acknowledgments

We thank all ED and ICU staff of participating hospitals for their support with this study. We also thank the staff of the Faculty of Public Health at Thai Binh University of Medicine and Pharmacy for their support and statistical advice. Finally, we thank Miss Phuong Thi Tran from the Center for Emergency Medicine of Bach Mai Hospital, Hanoi, Vietnam, and Miss

Truc-Cam Nguyen from Stanford University, Stanford, California, the United States of America, for their support with this study.

## Notes

Son Ngoc Do, Chinh Quoc Luong, My Ha Nguyen, et al. Predictive validity of the quick Sequential Organ Failure Assessment (qSOFA) score for the mortality in patients with sepsis in Vietnamese intensive care units. Abstract presented at European Society of Intensive Care Medicine (ESICM) LIVES 40 Conference; May 12th to 14th, 2022, 2022; Madrid, Spain.

## Author Contributions

**Conceptualization:** Son Ngoc Do, Chinh Quoc Luong.

**Data curation:** Son Ngoc Do, Chinh Quoc Luong, Nga Thi Nguyen, Dai Quang Huynh, Quoc Trong Ai Hoang, Co Xuan Dao, Thang Dinh Vu, Ha Nhat Bui, Hung Tan Nguyen, Hai Bui Hoang, Thuy Thi Phuong Le, Lien Thi Bao Nguyen, Phuoc Thien Duong, Tuan Dang Nguyen, Vuong Hung Le, Giang Thi Tra Pham.

**Formal analysis:** Chinh Quoc Luong, My Ha Nguyen, Dung Thi Pham.

**Investigation:** Son Ngoc Do, Chinh Quoc Luong, My Ha Nguyen, Dung Thi Pham, Nga Thi Nguyen, Dai Quang Huynh, Quoc Trong Ai Hoang, Co Xuan Dao, Thang Dinh Vu, Ha Nhat Bui, Hung Tan Nguyen, Hai Bui Hoang, Thuy Thi Phuong Le, Lien Thi Bao Nguyen, Phuoc Thien Duong, Tuan Dang Nguyen, Vuong Hung Le, Giang Thi Tra Pham, Tam Van Bui, Giang Thi Huong Bui, Jason Phua, Andrew Li, Thao Thi Ngoc Pham, Chi Van Nguyen, Anh Dat Nguyen.

**Methodology:** Son Ngoc Do, Chinh Quoc Luong, My Ha Nguyen, Dung Thi Pham, Co Xuan Dao, Jason Phua, Andrew Li.

**Project administration:** Son Ngoc Do, Chinh Quoc Luong.

**Supervision:** Son Ngoc Do, Chinh Quoc Luong, Co Xuan Dao, Jason Phua, Andrew Li.

**Writing – original draft:** Chinh Quoc Luong.

**Writing – review & editing:** Son Ngoc Do, My Ha Nguyen, Dung Thi Pham, Nga Thi Nguyen, Dai Quang Huynh, Quoc Trong Ai Hoang, Co Xuan Dao, Thang Dinh Vu, Ha Nhat Bui, Hung Tan Nguyen, Hai Bui Hoang, Thuy Thi Phuong Le, Lien Thi Bao Nguyen, Phuoc Thien Duong, Tuan Dang Nguyen, Vuong Hung Le, Giang Thi Tra Pham, Tam Van Bui, Giang Thi Huong Bui, Jason Phua, Andrew Li, Thao Thi Ngoc Pham, Chi Van Nguyen, Anh Dat Nguyen.

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
