## [Decision Letter · Decision Letter 0]

6 Jun 2022

PONE-D-21-40686Predictive validity of the quick Sequential Organ Failure Assessment (qSOFA) score for the mortality in patients with sepsis in Vietnamese intensive care unitsPLOS ONE

Dear Dr. Luong,

Thank you for submitting your manuscript to PLOS ONE. After careful consideration, we feel that it has merit but does not fully meet PLOS ONE’s publication criteria as it currently stands. Therefore, we invite you to submit a revised version of the manuscript that addresses the points raised during the review process.

We look forward to receiving your revised manuscript.

Kind regards,

Dinh-Toi Chu, PhD

Academic Editor

PLOS ONE

Journal Requirements:

2. Please amend your authorship list in your manuscript file to include author Andrew Li.

3. Thank you for submitting the above manuscript to PLOS ONE. During our internal evaluation of the manuscript, we found significant text overlap between your submission and the following previously published works, some of which you are an author.

- https://www.nature.com/articles/s41598-021-98165-8?code=7a678f24-643b-46d4-a84a-8f56197699e8&error=cookies_not_supported

- https://jamanetwork.com/journals/jama/fullarticle/2681801

Please revise the manuscript to rephrase the duplicated text, cite your sources, and provide details as to how the current manuscript advances on previous work. Please note that further consideration is dependent on the submission of a manuscript that addresses these concerns about the overlap in text with published work.

Reviewers' comments:

Reviewer #1: In the manuscript „Predictive validity of the quick Sequential Organ Failure Assessment (qSOFA) score for the mortality in patients with sepsis in Vietnamese intensive care units“, Do and colleagues assess the validity of the qSOFA score for the prediction of ICU and hospital mortality in Vietnamese intensive care units. The authors conclude – in line with several other authors – that the qSOFA score lack discriminatory power. However, in the study population, a qSOFA score value of three might be an indicator for ICU/hospital mortality.

Although age was not included in the final regression models, age might contribute to death. Did the authors consider an age criterion additionally to the qSOFA score for the identification of patients at risk? In patients with community-acquired pneumonia, we (https://doi.org/10.1016/j.cmi.2020.10.008) added the age criterion >= 65 years to the qSOFA score and compared this extended score to the CRB-65 criteria. After extension, the performance of the CRB-65 and the “qSOFA-65” criteria were – at least in the studied population – similar.

Overall, there remain issues to be solved in the main part and in the supplement of the manuscript.

Comments to the main part of the manuscript:

1) Please clarify throughout the manuscript whether multivariate analyses (several dependent variables) or multivariable analyses (several independent variables) were applied. This also applies to the supplement.

2) Abstract

a. Please add in the method paragraph, that a cut-off value for the qSOFA score was estimated based on the ROC curve analysis. Furthermore, please clarify (e.g. through re-ordering) in lines 87-89 that sensitivity and specificity correspond to the cutoff value and the p-value probably to the AUROC.

b. Please clarify which survival was assessed by the Kaplan-Meier curves and the log-rank test. From the results, it was 60-day (all-cause?) mortality.

c. Please clarify “on the four days representing the different seasons of 2019”. It might help to use “point prevalence study”, which is introduced in the method section of the main text.

3) Methods:

a. Please provide information on study registration.

b. Please clarify “mortality”. All-cause mortality?

c. There was a follow-up of 90 days but no mortality besides ICU and hospital mortality was defined as outcome. In the results, 60 days are considered. Please include this analysis in the method section.

d. Line 157/158: Please clarify “we used only data from Vietnam”. Probably, the authors only included hospitals in Vietnam. Were all Vietnamese hospitals from MOSAICS II included? Is somewhere a list of these participating hospitals available?

e. Please consider providing the dates (and the note on the seasons) in the study design section in the context of the point prevalence study. Then, only a short remark is needed in lines 163/164 to the selected days, which would facilitate reading. Furthermore, please consider in general providing the dates of the four days first and then highlight that these four days represented the different seasons.

f. Lines 165-167: In the main text, the authors state that patients were included if the patient was admitted due to sepsis and stays completely on one of the four pre-specified days. In Figure S1, it seems that patients can also be admitted or discharged on one of the four days to be included. Please clarify.

g. Lines 171/172: Please provide a reference to the subsequent section for the “common variables”. Furthermore, do they comprise all variables from the “dictionary”?

h. Line 197: Please clarify, whether IQR or the quartiles (Q1, Q3) are provided. The interquartile range is per definition the difference between Q3 and Q1. If necessary, please adapt the result section.

i. Line 197: What was the criterion for the decision between median and mean? Please add. Did the authors consider only relying on median with quartiles?

j. Lines 203-207: Please provide information on the approach to identify the cut-off value for the qSOFA score.

k. Lines 203-207: Was the performance comparable to the originally suggested qSOFA score cut-off of >=2 (doi:10.1001/jama.2016.0287)? Please provide information on this issue.

l. Lines 208-221: Did the authors account for possible centre effects in the logistic regression models?

m. Lines 208-221: Please consider not to provide links to results (tables) in the method section, but describe instead the analysis approach completely in words.

n. Lines 208-211: Please describe more clearly the variable selection. Which variables were considered for bivariate analysis? Which variables were included for further modelling based on the bivariate analysis? Which kind of modelling was used for the further analysis? What are the “different ways” of variable selection?

o. Line 211: The authors state that they performed a “bivariate analysis”. Please clarify, whether there was more than one dependent variable modelled in one model. Otherwise, it is probably a univariable logistic regression model (one dependent and one independent variable).

p. Line 214: Could the authors please provide the clinically important variables separately? They were probably included in step 2 irrespective of the results from step 1 (see also above comment).

q. Lines 217/218: Please provide information on the backward elimination approach (e.g., applied criteria).

r. Line 221: Please add “(adjusted)” in front of “odds ratio” to include results from multivariable regression models.

s. Lines 222/223: Please add whether there were corrections for multiple testing.

4) Results (lines 257-266)

a. Please consider restructuring. Suggestion: first, AUC values; second, identified cut-off value; third, sensitivity and specificity; forth, regression (and other) results. This would facilitate reading.

b. Adding predictive values might result in a deeper insight.

c. Please state the implication of the identified cut-off value, i.e. a patient must be positive for all three criteria.

d. Please provide more information about the regression (selection process; selected models).

e. Please check, whether all supplemental tables and figures are references in the main text or in the tables/figures of the main part.

5) Discussion:

a. Please start the discussion with a short summary of the study and the main findings.

b. Please introduce HIC, LMIC and ED as abbreviations.

c. Please consider avoiding figure and table references in the discussion. All figures and tables should be mentioned in the result section and no additional findings should be introduced in the discussion.

d. Line 322: Please clarify “respectively”. Can it be deleted?

e. Line 326: “Despite” instead of “Although”?

f. Lines 343/344: Please check the grammar and rephrase.

6) Tables

a. Overall

i. Please use “(Q1, Q3)” instead of “(IQR)”, because the interquartile range is per definition the difference between Q3 and Q1, i.e. the third and the first quartile.

ii. Please consider to provide consistently median with Q1 and Q3 instead of mean with SD for some characteristics.

iii. Please always provide the relative frequency with absolute counts, even in case of zero counts.

iv. Please indicate whether a p-value for a categorical variable was derived from the chi2 or the exact Fisher test.

b. Table 1:

i. Please add in the title the keyword “hospital mortality” – similar to the way in table 2.

ii. Please clarify the difference between “>0.999” and “1.000” in the p-value column (see e.g. supplemental tables).

iii. Please add the definition of septic shock in the variable section within the method section.

iv. Please consider “Documented comorbidities” instead of “Comorbidities”.

c. Table 2:

i. The authors write “Respiratory support, no. (%) and median (IQR), days”, but I could not identify any median in this context. Are, e.g., the number of days with mechanical ventilation missing? Please check.

ii. Please introduce the formats in this table. In case of missing information, the authors provide n/N (%), which should be stated somewhere. If the number of patients with information is constant across a section, the section heading might receive a footnote with the information on the number of patients with missing information.

iii. Missing abbreviation: “no.”

d. Table 3:

i. Please check the use of “bivariate” and “multivariate” (versus “univariable” and “multivariable”; see comments above).

ii. Please consider “Documented comorbidities” instead of “Comorbidities”.

iii. Please add a short description of the variable selection.

iv. Suggestion: “qSOFA score of 3” instead of “qSOFA” to shorten the table.

7) Figures:

a. Overall:

i. Please enlarge the numbers on the axes and the legend text.

ii. Please consider highlighting the abbreviations by introducing the word “abbreviation”.

iii. The resolution of the figures is too low. If this was not caused by the submission system, please adapt.

b. Figure 1:

i. Please check for consistent format with ROC curves in the supplement (S2-S4).

ii. Please clarify that the cutoff value of three corresponds to sensitivity and specificity and the p-value probably to the AUROC.

c. Figure 2:

i. Please consider the inclusion of the observations / results (currently provided in the description) into the main text.

ii. Please clarify the meaning of “, 2019” in the first sentence of the description.

iii. Description of the line types might be deleted from the description as the information is already provided through the legend.

iv. Please provide information on censoring.

v. Please provide the number of patients at risk for the qSOFA score groups below the time axis.

vi. Please consider re-naming the time axis in “Days since …” with “...” naming the time origin.

vii. Please avoid abbreviations in the axes names.

Additional comments to the supplementary results of the manuscript:

1) Overall:

a. Please do not mix figures and tables, i.e. please provide first the figures and then the tables (or vice versa).

b. Please introduce a list of tables and figures with page numbers in the beginning of the document.

2) Figures:

a. Overall: Please consider highlighting the abbreviations by introducing the word “abbreviation”.

b. Figure S1: Is it possible to provide a number of screened patients (first box) and number of excluded patients (overall and stratified by reason)? If possible, could the authors please provide these numbers?

c. Figures S2-S5:

i. Please consider a similar format for all ROC curves (including legend, title, note, abbreviations [e.g., AUC versus AUROC]).

ii. Are figures S2 and S3 as well as S4 and S5 needed?

iii. S2+S4: Please clarify that the cutoff value of three corresponds to sensitivity and specificity and the p-value probably to the AUROC.

iv. S3+S5: Please clarify “Area Under the Curve” and “Coordinates of the Curve” below the description.

d. Figure S6:

i. Please introduce a space between “qSOFA” and “group” in the legend.

ii. Please consider avoiding the line description in the description, because of the provided legend.

iii. Please provide information on censoring, if present.

iv. Please provide the number of patients at risk for the qSOFA score groups below the time axis.

v. Please consider re-naming the time axis in “Days since …” with “...” naming the time origin.

vi. Please avoid abbreviations in the axes names.

vii. Please remove the title “Survival Function” above the plot.

3) Tables:

a. Overall:

i. Please provide all abbreviations in the description.

ii. Please consider to write “p-value” instead of “p”.

iii. Please use “(Q1, Q3)” instead of “(IQR)”, because the interquartile range is per definition the difference between Q3 and Q1, i.e. the third and the first quartile.

iv. Please consider to provide consistently median with Q1 and Q3 instead of mean with SD for some characteristics.

v. Please always provide the relative frequency with absolute counts, even in case of zero counts.

vi. Please indicate whether a p-value for a categorical variable was derived from the chi2 or the exact Fisher test.

b. Table S1:

i. Please consider “Documented comorbidities” instead of “Comorbidities”.

ii. “SIRS” probably corresponds to the “SIRS criteria”. Please clarify.

iii. Similarly, please consider adding the word “score” to “SOFA” and “qSOFA”.

iv. Regarding missing information/values, were all characteristics documented in all patients? If not, please provide the number of patients with missing information for a specific characteristic.

c. Table S2:

i. Line 3, first column: Probably “median (IQR) days” instead of “median (IQR), days” and “no. (%) of patients” instead of “no. (%)” to avoid misinterpretation.

ii. Please introduce the formats in this table. In case of missing information, the authors provide n/N (%), which should be stated somewhere. If the number of patients with information is constant across a section, the section heading might receive a footnote with the information on the number of patients with missing information.

d. Table S3:

i. See comments for Table S1.

ii. Please clarify “-“ in case of counts.

e. Table S4: See comments for Table S2.

f. Table S5:

i. See comments for Table S1.

ii. “bivariate regression analyses” indicates univariable logistic regression modelling? Please clarify.

iii. Please state the meaning of “-“.

iv. Please clarify “Frequency”. Please provide the number of patients included in the respective model.

v. Please clarify the difference between “>0.999” and “1.000” in the p-value column.

g. Table S6:

i. Against the provided table, the authors included several independent variables in the model and performed a “multivariable logistic regression analyses” instead of a “multivariate logistic regression analyses” (several dependent variables). Please clarify.

ii. Please clarify “unit”. I would expect that individual level data was used and that for each variable the patient exhibited a specific category.

iii. Please mark the final model.

h. Table S7: See comments for Table S5.

i. Table S8: See comments for Table S6.

Reviewer #2: I would like to thank all authors for their wonderful work. It is very useful study. Mainly in developing countries managing sepsis.

Minor comment:

1. In the discussion section please add few possible reasons, why SOFA score does not show higher prediction (Based on the ROC) in the selected sample.

Reviewer #3: No suggestions. The manuscript is well written and can be accept in its present state. It is just advised the article is read and re-read to avoid any typos. A few typo errors were detected so a through proof reading is advised.

Other than that no suggestions are provided for improvement.

Reviewer #4: Reviewer comments

Manuscript Number: PONE-D-21-40686

Title "Predictive validity of the quick Sequential Organ Failure Assessment (qSOFA) score for the mortality in patients with sepsis in Vietnamese intensive care units".

Generally speaking:

Thank you for providing me the opportunity to review this manuscript that raises important and interesting issues about predictive validity of the quick Sequential Organ Failure Assessment (qSOFA) score for the mortality in patients with sepsis in intensive care units in one of the developing countries.

Comment 1:

1. ABSTRACT:

a) No need for details of methodology in the abstract. Only main points. It should not include the type of statistical analysis.

b) Methodology has to mention the elements included as measuring parameter but without details.

c) Abbreviations as APACHE, MOSAICS, and SIRS must be defined upon first appearance in the keywords.

d) It is preferable that the keywords should be 5-6 words only.

Comment 2:

2. INTRODUCTION:

a) Global/ Regional/ Vietnam prevalence of the mortality of ICU patients with sepsis should be mentioned.

b) Factors associated with mortality in patients with sepsis upon Vietnamese intensive care unit admission should be clearly stated.

c) Explaining why this topic was chosen for analysis in this article is not well written. The benefits of conducting the study to the community should be explained.

Comment 3:

3. METHODS:

a) Were the samples normal or not?

b) Tests of significant for each type of variables should be mentioned.

c) In lines 158 and 159, why predominantly neurosurgical, coronary, and cardiothoracic ICUs were excluding?

d) The basis of sample size calculation should be clearly stated to know the confidence level and the margin of error.

Comment 4:

4. RESULTS:

a) No need to conduct bivariate logistic regression because it doesn’t consider the confounding variables.

b) What was the criteria to pass your variables from bivariate to multivariate logistic regression?

Comment 5:

5. DISCUSSION:

a) It is advisable to explain the study objective at the beginning of the discussion.

b) Compare the findings of the study with other findings and state the reasons for the strengths and weaknesses in each section. The manuscript could be greatly strengthened if the authors could provide highlight on the main and significant factors associated with hospital and ICU mortalities in patients with sepsis in other developing and developed countries with similar context.

c) The manuscript could also be greatly strengthened if the authors could provide highlight on the main and significant life-sustaining treatments during ICU stay and outcomes of ICU patients with sepsis according to hospital survivability in other developing and developed countries with similar context.

d) In brief, discuss by using the scientific reasoning the differences in accuracy of the qSOFA score to predict the risk of hospital and ICU deaths from sepsis.

e) In lines 305 and 306, the sentence “Our 304 figures for the hospital and ICU mortality rates are in line with the figures reported in the 305 worldwide Intensive Care over Nations (ICON) study (19.3% [26/439] to 47.2% [17/141], and 11.9% [16/439] to 39.5 [15/141], respectively)” was not clearly stated.

f) Abbreviations as HIC must be defined upon first appearance in the discussion.

Comment 6:

6. CONCLUSION:

Please write suggestions for improvement.

Comment 7:

7. STRENGTHS AND LIMITATIONS:

Strengths and limitations of the study should be analyzed in separate paragraphs.

---

## [Author Response · Author response to Decision Letter 0]

1 Aug 2022

Professor Dinh-Toi Chu, PhD.

Academic Editor

PLOS ONE

August 1, 2022

Dear Prof. Dinh-Toi Chu,

On behalf of all authors, I am resubmitting herewith our revised manuscript entitled “Predictive validity of the quick Sequential Organ Failure Assessment (qSOFA) score for the mortality in patients with sepsis in Vietnamese intensive care units” (PONE-D-21-40686).

We sincerely appreciate the kind comments and points raised by the Editors and by the Reviewers. We have carefully considered all comments and suggestions and revised our manuscript following each of these points. These comments have enabled us to substantially improve our manuscript. We hope that Editor will find our revised manuscript suitable for publication in PLOS ONE.

We confirm that this work is original and has not been published elsewhere nor is it currently under consideration for publication elsewhere. All authors have read, approved the manuscript, and agreed to authorship and order of authorship for this manuscript, and all authors have the appropriate permissions and rights to the reported data.

We have provided our point-by-point responses to the comments of the Editors and the Reviewers below.

We thank you for your kind consideration of this submission.

Sincerely yours,

Chinh Quoc Luong, MD., PhD.

Center for Emergency Medicine,

Bach Mai Hospital,

No. 78, Giai Phong, Phuong Mai ward, Dong Da district, Hanoi 100000, Vietnam

Email: luongquocchinh@gmail.com

We thank the Editors and the Reviewers for the valuable comments and suggestions that greatly helped us to improve the contents of this paper. In what follows, we will use the boldface to indicate comments from the Editors and the Reviewers, the standard font face for our responses and we highlighted in yellow the modifications that we did to the manuscript.

RESPONSE TO EDITORS

Thank you for submitting your manuscript to PLOS ONE. After careful consideration, we feel that it has merit but does not fully meet PLOS ONE’s publication criteria as it currently stands. Therefore, we invite you to submit a revised version of the manuscript that addresses the points raised during the review process.

Our answer:

We thank you for the positive feedback. We have carefully considered the Reviewers' comments and suggestions and have revised our manuscript following each of these points.

Thank you for this comment. We have submitted the revised manuscript on time.

Our answer:

Thank you for this comment. We have included a rebuttal letter, a marked-up copy of the manuscript, and an unmarked version of the revised manuscript when submitting our revised manuscript.

Our answer:

Thank you for this comment. We do not have any changes in our financial disclosure.

Our answer:

Thank you for this comment. Laboratory protocol does not apply to our study, but study protocol does. Our study protocol has included in the Methods section.

Journal Requirements:

1. Please ensure that your manuscript meets PLOS ONE's style requirements, including those for file naming. The PLOS ONE style templates can be found at:

Our answer:

We have ensured that our manuscript meets PLOS ONE's style requirements, including those for file naming.

2. Please amend your authorship list in your manuscript file to include author Andrew Li.

Our answer:

Thank you for this comment. We have amended our authorship list to include author Andrew Li in our manuscript file.

3. Thank you for submitting the above manuscript to PLOS ONE. During our internal evaluation of the manuscript, we found significant text overlap between your submission and the following previously published works, some of which you are an author.

https://www.nature.com/articles/s41598-021-98165-8?code=7a678f24-643b-46d4-a84a-8f56197699e8&error=cookies_not_supported

https://jamanetwork.com/journals/jama/fullarticle/2681801

Please revise the manuscript to rephrase the duplicated text, cite your sources, and provide details as to how the current manuscript advances on previous work. Please note that further consideration is dependent on the submission of a manuscript that addresses these concerns about the overlap in text with published work.

Our answer:

Thank you for pointing this out. We have revised the manuscript to rephrase the duplicated text, cite our sources, and provide details of how the current manuscript advances on previous work.

RESPONSE TO REVIEWERS

Reviewers' comments:

Reviewer #1

In the manuscript “Predictive validity of the quick Sequential Organ Failure Assessment (qSOFA) score for the mortality in patients with sepsis in Vietnamese intensive care units”, Do and colleagues assess the validity of the qSOFA score for the prediction of ICU and hospital mortality in Vietnamese intensive care units. The authors conclude – in line with several other authors – that the qSOFA score lack discriminatory power. However, in the study population, a qSOFA score value of three might be an indicator for ICU/hospital mortality.

Our answer:

Thank the Reviewer so much for taking the time to leave his/her excellent reviews.

Although age was not included in the final regression models, age might contribute to death. Did the authors consider an age criterion additionally to the qSOFA score for the identification of patients at risk? In patients with community-acquired pneumonia, we (https://doi.org/10.1016/j.cmi.2020.10.008) added the age criterion >= 65 years to the qSOFA score and compared this extended score to the CRB-65 criteria. After extension, the performance of the CRB-65 and the “qSOFA-65” criteria were – at least in the studied population – similar.

Our answer:

Thank you for this valuable comment. In the previous version of the manuscript, we included the age by the decades as a categorical variable (< 20; 20 - 39; 40 - 59; and ≥ 60 years) in both binary and multivariate logistic regression models. However, we did not find any independent association between age and death. Additionally, we have included the age ≥65 years in these models and we also did not find any independent association between the age ≥65 years and death, as shown in Tables S5, S6, S7, and S8 (Additional file 2).

We have added the age criterion ≥65 years to the qSOFA score and compared this extended score (qSOFA-65) to the original score (qSOFA). However, these scores both had a poor discriminatory ability for the prognosis of intensive care unit patients with sepsis (Fig S3 and S4 as shown in Additional file 2). These findings differ from the results of your published paper,[1] which might be due to a relatively small sample size of our cohort and the different population between the two studies. We have added these results to the Results section (Lines 284-289) and have provided a brief discussion in the Discussion section (Lines 356-361). 

[1] Kesselmeier M, Pletz MW, Blankenstein AL, Scherag A, Bauer T, Ewig S, Kolditz M. Validation of the qSOFA score compared to the CRB-65 score for risk prediction in community-acquired pneumonia. Clin Microbiol Infect. 2021 Sep;27(9):1345.e1-1345.e6. doi: 10.1016/j.cmi.2020.10.008. Epub 2020 Oct 10. PMID: 33049414.

Overall, there remain issues to be solved in the main part and in the supplement of the manuscript.

Our answer:

Thank the Reviewer so much for taking the time to leave the excellent reviews. We have carefully considered the Reviewers' comments and suggestions and revised our submission following each of these comments.

Comments to the main part of the manuscript:

1) Please clarify throughout the manuscript whether multivariate analyses (several dependent variables) or multivariable analyses (several independent variables) were applied. This also applies to the supplement.

Our answer:

Thank you for pointing this out. We have clarified throughout the manuscript whether multivariate analyses (several dependent variables) or multivariable analyses (several independent variables) were applied.

2) Abstract

a. Please add in the method paragraph, that a cut-off value for the qSOFA score was estimated based on the ROC curve analysis. Furthermore, please clarify (e.g. through re-ordering) in lines 87-89 that sensitivity and specificity correspond to the cutoff value and the p-value probably to the AUROC.

Our answer:

Thank you for this comment. We have added a paragraph to the method section where we described how to estimate the cut-off value (Lines 80-81). We have also clarified the sensitivity and specificity corresponding to the cut-off value and the p-value corresponding to the AUROC (Lines 85-87).

b. Please clarify which survival was assessed by the Kaplan-Meier curves and the log-rank test. From the results, it was 60-day (all-cause?) mortality.

Our answer:

Thank you for pointing this out. We also thank you for this valuable comment which helps us find out the mistakes in the analysis and interpretation of data for the work, as follows:

In the present study, we followed all patients till hospital discharge, death in the intensive care unit (ICU)/hospital, and up to 90-day post-enrollment, whichever was the earliest. In our study, data on the time of 60- and 90-day post-enrollment was not available. However, we assumed patients who survived to discharge from the hospital before the time of 60-day post-enrollment were still alive after the time of 60-day post-enrollment, and based on this cut-off time point for using the Kaplan-Meier time-to-event analysis to estimate the survival functions and the log-rank test to compare survival by the qSOFA score groups. This interpretation was a mistake. Therefore, we have removed the Kaplan-Meier time-to-event analysis and its relevant results from the manuscript and supplementary file.

In our study, both ICU and hospital mortalities were from all causes. Therefore, we have defined ICU and hospital mortalities in the Methods section of the Abstract (Lines 77-78) and the Outcomes section of the main text (Lines 195-197).

c. Please clarify “on the four days representing the different seasons of 2019”. It might help to use “point prevalence study”, which is introduced in the method section of the main text.

Our answer:

Thank you for pointing this out. We have clarified this statement in the methods of the abstract (Lines 76-77).

3) Methods:

a. Please provide information on study registration.

Our answer:

Thank you for this comment. The present study is part of the Management of Severe sepsis in Asia’s Intensive Care unitS II (MOSAICS II) study,[1] with a study registration number (CTRI/2019/01/016898) on the Clinical trials registry – India website.[2] We have provided information on study registration (Line 97).

[1] Li A, Ling L, Qin H, Arabi YM, Myatra SN, Egi M, Kim JH, Mat Nor MB, Son DN, Fang WF, Wahyuprajitno B, Hashmi M, Faruq MO, Patjanasoontorn B, Al Bahrani MJ, Shrestha BR, Shrestha U, Nafees KMK, Sann KK, Palo JEM, Mendsaikhan N, Konkayev A, Detleuxay K, Chan YH, Du B, Divatia JV, Koh Y, Gomersall CD, Phua J; MOSAICS II Study Group, for the Asian Critical Care Clinical Trials Group. Epidemiology, Management, and Outcomes of Sepsis in Intensive Care Units Among Countries of Differing National Wealth Across Asia. Am J Respir Crit Care Med. 2022 Jun 28. doi: 10.1164/rccm.202112-2743OC. Epub ahead of print. PMID: 35763381.

[2] http://ctri.nic.in/Clinicaltrials/login.php

b. Please clarify “mortality”. All-cause mortality?

Our answer:

In our study, both ICU and hospital mortalities were from all causes. Therefore, we have defined ICU and hospital mortalities in the Outcomes section of the Methods section (Lines 195-197).

c. There was a follow-up of 90 days but no mortality besides ICU and hospital mortality was defined as outcome. In the results, 60 days are considered. Please include this analysis in the method section.

Our answer:

Thank you for this comment. In the Outcomes section, we defined hospital mortality as a primary outcome (Lines 195-197). We also described intensive care unit mortality as a secondary outcome (Lines 195-197). In our study, we followed all patients till hospital discharge, death in the intensive care unit (ICU)/hospital, and up to 90-day post-enrollment, whichever was the earliest. However, we could not evaluate the 60- and 90-day mortalities because data on the time point of 60- and 90-day post-enrollment was unavailable.

d. Line 157/158: Please clarify “we used only data from Vietnam”. Probably, the authors only included hospitals in Vietnam. Were all Vietnamese hospitals from MOSAICS II included? Is somewhere a list of these participating hospitals available?

Our answer:

Thank you for this comment. The present study is part of the Management of Severe sepsis in Asia’s Intensive Care unitS II (MOSAICS II) study,[1] which collects data on the management of sepsis in Asia, including Vietnam. The present study used only data from Vietnam. A total of 15 adult ICUs participated in the MOSAICS II study from 14 hospitals, of which five are the central hospital and nine are provincial, district, or private hospitals, throughout Vietnam. We have provided the list of these participating hospitals in Tables S1 and S3 as shown in Additional file 2.

[1] Li A, Ling L, Qin H, Arabi YM, Myatra SN, Egi M, Kim JH, Mat Nor MB, Son DN, Fang WF, Wahyuprajitno B, Hashmi M, Faruq MO, Patjanasoontorn B, Al Bahrani MJ, Shrestha BR, Shrestha U, Nafees KMK, Sann KK, Palo JEM, Mendsaikhan N, Konkayev A, Detleuxay K, Chan YH, Du B, Divatia JV, Koh Y, Gomersall CD, Phua J; MOSAICS II Study Group, for the Asian Critical Care Clinical Trials Group. Epidemiology, Management, and Outcomes of Sepsis in Intensive Care Units Among Countries of Differing National Wealth Across Asia. Am J Respir Crit Care Med. 2022 Jun 28. doi: 10.1164/rccm.202112-2743OC. Epub ahead of print. PMID: 35763381.

e. Please consider providing the dates (and the note on the seasons) in the study design section in the context of the point prevalence study. Then, only a short remark is needed in lines 163/164 to the selected days, which would facilitate reading. Furthermore, please consider in general providing the dates of the four days first and then highlight that these four days represented the different seasons.

Our answer:

Thank you for this comment. We have provided the dates (and the note on the seasons) in the Study design and setting section (Lines 155-156). We have also provided the dates of the four days first and highlighted that these four days represented the different seasons in the Participants section (Lines 162-163).

f. Lines 165-167: In the main text, the authors state that patients were included if the patient was admitted due to sepsis and stays completely on one of the four pre-specified days. In Figure S1, it seems that patients can also be admitted or discharged on one of the four days to be included. Please clarify.

Our answer:

Thank you for pointing this out. In our study, we included all patients, aged ≥18 years old, who were admitted to the intensive care units (ICUs) for sepsis, and who were still in the ICUs from 00:00 hour to 23:59 hour of the study days (Figure S1 as shown in Additional file 2). However, Figure S1 shows that two patients were admitted on the day of patient enrollment for sepsis and died/discharged on the same day, of which "discharged" (also meaning "discharged to die") was defined as the patients were in grave condition or dying and were classified with a death in the ICU at the time of discharge. We have clarified this definition in Figure S1 as shown in Additional file 2.

g. Lines 171/172: Please provide a reference to the subsequent section for the “common variables”. Furthermore, do they comprise all variables from the “dictionary”?

Our answer:

Thank you for pointing this out. We have provided a reference to where the common variables to be able to find (Lines 169-170), of which the data collection form comprises all variables from the dictionary.

h. Line 197: Please clarify, whether IQR or the quartiles (Q1, Q3) are provided. The interquartile range is per definition the difference between Q3 and Q1. If necessary, please adapt the result section.

Our answer:

Thank you for pointing this out. We reported data as the medians and quartiles (Q1-Q3) instead of the medians and interquartile ranges (IQRs) in the case of non-normal distribution for continuous variables throughout the manuscript. Therefore, this error is a typing mistake. We have corrected the error. We have also adapted it in the Results section.

i. Line 197: What was the criterion for the decision between median and mean? Please add. Did the authors consider only relying on median with quartiles?

Our answer:

Thank you for pointing this out. We have added the criterion for the decision between medians and means in the Statistical analyses section (Lines 213-215). Yes, we did. We considered only relying on the medians with quartiles instead of the medians with interquartile ranges.

j. Lines 203-207: Please provide information on the approach to identify the cut-off value for the qSOFA score.

Our answer:

Thank you for pointing this out. We have provided information on the approach to identify the cut-off value for the qSOFA score (Lines 225-226).

k. Lines 203-207: Was the performance comparable to the originally suggested qSOFA score cut-off of >=2 (doi:10.1001/jama.2016.0287)? Please provide information on this issue.

Our answer:

Thank you for this comment. The present study shows that the predictive validity for hospital mortality of qSOFA score above baseline risk (AUROC=0.610; 95% CI: 0.538-0.681; cut-off value: ≥2.5) was similar to those of the originally-suggested qSOFA score (AUROC = 0.66; 95% CI, 0.64-0.68; cutoff value: ≥2) in patients with sepsis in the ICU.[1],[2] We have added this sentence in the Discussion section (Lines 353-356).

[1] Seymour CW, Liu VX, Iwashyna TJ, Brunkhorst FM, Rea TD, Scherag A, Rubenfeld G, Kahn JM, Shankar-Hari M, Singer M, Deutschman CS, Escobar GJ, Angus DC. Assessment of Clinical Criteria for Sepsis: For the Third International Consensus Definitions for Sepsis and Septic Shock (Sepsis-3). JAMA. 2016 Feb 23;315(8):762-74. doi: 10.1001/jama.2016.0288. Erratum in: JAMA. 2016 May 24-31;315(20):2237. PMID: 26903335; PMCID: PMC5433435.

[2] Singer M, Deutschman CS, Seymour CW, Shankar-Hari M, Annane D, Bauer M, Bellomo R, Bernard GR, Chiche JD, Coopersmith CM, Hotchkiss RS, Levy MM, Marshall JC, Martin GS, Opal SM, Rubenfeld GD, van der Poll T, Vincent JL, Angus DC. The Third International Consensus Definitions for Sepsis and Septic Shock (Sepsis-3). JAMA. 2016 Feb 23;315(8):801-10. doi: 10.1001/jama.2016.0287. PMID: 26903338; PMCID: PMC4968574.

l. Lines 208-221: Did the authors account for possible centre effects in the logistic regression models?

Our answer:

Thank you for this valuable comment. We did not account for possible central hospital effects on the logistic regression models. However, we have added this variable to the multivariate logistic regression models and found that both the central hospital (adjusted OR: 1.992; 95% CI: 1.110-3.573; p=0.021) and qSOFA score of 3 (adjusted OR: 3.358; 95% CI: 1.756-6.422; p <0.001) were independently associated with the increased risk of deaths in the hospital (Table 3). In contrast to the qSOFA score of 3 (adjusted OR: 3.060; 95% CI: 1.651-5.671; p <0.001), the central hospital was not independently associated with the increased risk of deaths in the ICU (Table 3).

m. Lines 208-221: Please consider not to provide links to results (tables) in the method section, but describe instead the analysis approach completely in words.

Our answer:

Thank you for this comment. Thank you for this comment. We have removed links to results (tables) in the Method section, and we have described the analysis approach completely in words.

n. Lines 208-211: Please describe more clearly the variable selection. Which variables were considered for bivariate analysis? Which variables were included for further modelling based on the bivariate analysis? Which kind of modelling was used for the further analysis? What are the “different ways” of variable selection?

Our answer:

Thank you for this comment. We have described more clearly the variable selection and answered the Reviewer's questions (Lines 224-241).

o. Line 211: The authors state that they performed a “bivariate analysis”. Please clarify, whether there was more than one dependent variable modelled in one model. Otherwise, it is probably a univariable logistic regression model (one dependent and one independent variable).

Our answer:

Thank you for pointing this out. We had some typing mistakes in terms of logistic regression models. We performed the binary logistic regression analysis instead of the bivariate/univariable logistic regression analysis. We have clarified these issues throughout the manuscript.

p. Line 214: Could the authors please provide the clinically important variables separately? They were probably included in step 2 irrespective of the results from step 1 (see also above comment).

Our answer:

Thank you for this comment. We have provided the clinically important variables separately (Line 230). However, we used a stepwise backward elimination method to select variables for the multivariate logistic regression analysis, so these variables were not probably included in step 2 if there were non-significant contributions to the outcome from step 1.

q. Lines 217/218: Please provide information on the backward elimination approach (e.g., applied criteria).

Our answer:

Thank you for this comment. We have provided information on the backward elimination approach in the Methods section (Lines 234-239).

r. Line 221: Please add “(adjusted)” in front of “odds ratio” to include results from multivariable regression models.

Our answer:

Thank you for this comment. We have added the term “(adjusted)” in front of “odds ratios” in the multivariate logistic regression models.

s. Lines 222/223: Please add whether there were corrections for multiple testing.

Our answer:

Thank you for this comment. We have added whether there were corrections for multiple testing (Line 243).

4) Results (lines 257-266)

a. Please consider restructuring. Suggestion: first, AUC values; second, identified cut-off value; third, sensitivity and specificity; forth, regression (and other) results. This would facilitate reading.

Our answer:

Thank you for this comment. We have restructured the presentation of these results (Lines 279-289).

b. Adding predictive values might result in a deeper insight.

Our answer:

Thank you for this comment.

c. Please state the implication of the identified cut-off value, i.e. a patient must be positive for all three criteria.

Our answer:

Thank you for this comment. We have stated the implication of the identified cut-off value (i.e., a patient must be positive for all three criteria) (Lines 290-292).

d. Please provide more information about the regression (selection process; selected models).

Our answer:

Thank you for this comment. We have provided more information about the regression (selection process; selected models) in the Methods (Lines 224-241) and Results sections (Lines 292-298).

e. Please check, whether all supplemental tables and figures are references in the main text or in the tables/figures of the main part.

Our answer:

Thank you for this comment. We have cited all tables/figures of the main text of the manuscript and all supplemental tables/figures.

5) Discussion:

a. Please start the discussion with a short summary of the study and the main findings.

Our answer:

Thank you for this comment. We have started the discussion with a short summary of the study and the main findings (Lines 309-312).

b. Please introduce HIC, LMIC and ED as abbreviations.

Our answer:

Thank you for this comment. We have introduced HIC (Lines 322-323), LMIC (Line 114) and ED (Line 128) as abbreviations.

c. Please consider avoiding figure and table references in the discussion. All figures and tables should be mentioned in the result section and no additional findings should be introduced in the discussion.

Our answer:

Thank you for this comment. We have removed Figure and Table references in the Discussion section.

d. Line 322: Please clarify “respectively”. Can it be deleted?

Our answer:

Thank you for this comment. We have deleted “respectively”.

e. Line 326: “Despite” instead of “Although”?

Our answer:

Thank you for this comment. We have replaced "Although” with “Despite” (Line 324).

f. Lines 343/344: Please check the grammar and rephrase.

Our answer:

Thank you for this comment. We have checked the grammar and rephrased this sentence (Lines 370-371).

6) Tables

a. Overall

i. Please use “(Q1, Q3)” instead of “(IQR)”, because the interquartile range is per definition the difference between Q3 and Q1, i.e. the third and the first quartile.

Our answer:

Thank you for this comment. We have replaced “(IQR)” with “(Q1, Q3)”.

ii. Please consider to provide consistently median with Q1 and Q3 instead of mean with SD for some characteristics.

Our answer:

Thank you for this comment. We have provided the consistently median with Q1 and Q3 instead of the mean with SD for some characteristics.

iii. Please always provide the relative frequency with absolute counts, even in case of zero counts.

Our answer:

Thank you for this comment. We have provided the relative frequency with absolute counts, even in the case of zero counts.

iv. Please indicate whether a p-value for a categorical variable was derived from the chi2 or the exact Fisher test.

Our answer:

Thank you for this comment. We have indicated whether a p-value for a categorical variable was derived from the χ2 test or Fisher exact test.

b. Table 1:

i. Please add in the title the keyword “hospital mortality” – similar to the way in table 2.

Our answer:

Thank you for this comment. We have added in the title the keyword “hospital mortality” to Tables 1 and 2.

ii. Please clarify the difference between “>0.999” and “1.000” in the p-value column (see e.g. supplemental tables).

Our answer:

Thank you for this comment. The p-value is generally not found to be 1. It only becomes 1 when the values of both groups are the same (identical). However, when the p-value approaches near 1, e.g., 0.999 or 0.989, it is regarded as 1. Our study presented some p-values at >0.999 because they are near 1.

iii. Please add the definition of septic shock in the variable section within the method section.

Our answer:

Thank you for this comment. We have added the definition of septic shock in the Variables section within the Method section (Lines 187-189).

iv. Please consider “Documented comorbidities” instead of “Comorbidities”.

Our answer:

Thank you for this comment. We have replaced “Comorbidities” with “Documented comorbidities”.

c. Table 2:

i. The authors write “Respiratory support, no. (%) and median (IQR), days”, but I could not identify any median in this context. Are, e.g., the number of days with mechanical ventilation missing? Please check.

Our answer:

Thank you for pointing this out. This error is a typing mistake. We have deleted “and median (IQR), days”.

ii. Please introduce the formats in this table. In case of missing information, the authors provide n/N (%), which should be stated somewhere. If the number of patients with information is constant across a section, the section heading might receive a footnote with the information on the number of patients with missing information.

Our answer:

Thank you for this comment. We have reformatted this table as required by the Reviewer.

iii. Missing abbreviation: “no.”

Our answer:

Thank you for this comment. We have deleted "no."

d. Table 3:

i. Please check the use of “bivariate” and “multivariate” (versus “univariable” and “multivariable”; see comments above).

Our answer:

Thank you for pointing this out. We had some typing mistakes in terms of logistic regression models. We performed the binary and multivariate logistic regression analysis instead of the bivariate/univariable and multivariable logistic regression analysis. We have amended these mistakes throughout the manuscript and Additional file.

ii. Please consider “Documented comorbidities” instead of “Comorbidities”.

Our answer:

Thank you for this comment. We have replaced “Comorbidities” with “Documented comorbidities”.

iii. Please add a short description of the variable selection.

Our answer:

Thank you for this comment. We have added a short description of the variable selection in the footnote of Table 3.

iv. Suggestion: “qSOFA score of 3” instead of “qSOFA” to shorten the table.

Our answer:

Thank you for this comment. We have replaced “qSOFA” with "qSOFA score of 3”.

7) Figures:

a. Overall:

i. Please enlarge the numbers on the axes and the legend text.

Our answer:

Thank you for this comment. We have enlarged the numbers on the axes and the legend text.

ii. Please consider highlighting the abbreviations by introducing the word “abbreviation”.

Our answer:

Thank you for this comment. We have highlighted the abbreviations by introducing the word “abbreviation”.

iii. The resolution of the figures is too low. If this was not caused by the submission system, please adapt.

Our answer:

Thank you for this comment. We ensure that our figures have a resolution of 300 pixels per inch (PPI) and appear sharp, not pixelated, and we uploaded them separately to the submission system.

b. Figure 1:

i. Please check for consistent format with ROC curves in the supplement (S2-S4).

Our answer:

Thank you for this comment. We have provided Figures S2-S4 in an appropriate format.

ii. Please clarify that the cutoff value of three corresponds to sensitivity and specificity and the p-value probably to the AUROC.

Our answer:

Thank you for this comment. We have clarified that the cut-off value of three corresponds to sensitivity, specificity, and the p-value of the AUROCs.

c. Figure 2:

i. Please consider the inclusion of the observations / results (currently provided in the description) into the main text.

ii. Please clarify the meaning of “, 2019” in the first sentence of the description.

iii. Description of the line types might be deleted from the description as the information is already provided through the legend.

iv. Please provide information on censoring.

v. Please provide the number of patients at risk for the qSOFA score groups below the time axis.

vi. Please consider re-naming the time axis in “Days since …” with “...” naming the time origin.

vii. Please avoid abbreviations in the axes names.

Our answer:

Thank you for these comments. We have removed the Kaplan-Meier time-to-event analyses and the relevant results from the manuscript and supplementary file due to the mistakes mentioned in the comments above.

Additional comments to the supplementary results of the manuscript:

1) Overall:

a. Please do not mix figures and tables, i.e. please provide first the figures and then the tables (or vice versa).

Our answer:

Thank you for pointing this out. We have amended this issue. We have provided first the Figures and then the Tables.

b. Please introduce a list of tables and figures with page numbers in the beginning of the document.

Our answer:

Thank you for this comment. We have introduced a list of tables and figures with page numbers at the beginning of the document.

2) Figures:

a. Overall: Please consider highlighting the abbreviations by introducing the word “abbreviation”.

Our answer:

Thank you for this comment. We have highlighted the abbreviations by introducing the word “abbreviation”.

b. Figure S1: Is it possible to provide a number of screened patients (first box) and number of excluded patients (overall and stratified by reason)? If possible, could the authors please provide these numbers?

Our answer:

Thank you for this comment. We have provided the number of screened patients (first box) and the overall number of excluded patients. However, we did not have data on the excluded patients stratified by reason.

c. Figures S2-S5:

i. Please consider a similar format for all ROC curves (including legend, title, note, abbreviations [e.g., AUC versus AUROC]).

Our answer:

Thank you for this comment. We have provided a similar format for all ROC curves (including legend, title, note, and abbreviations [e.g., AUC vs. AUROC]).

ii. Are figures S2 and S3 as well as S4 and S5 needed?

Our answer:

Thank you for this comment. We have removed Figures S2, S3, S5, and S6 from the Additional file.

iii. S2+S4: Please clarify that the cutoff value of three corresponds to sensitivity and specificity and the p-value probably to the AUROC.

Our answer:

Thank you for this comment. We have removed Figures S2, S3, S5, and S6 from the Additional file. We have also clarified the cut-off value of three corresponds to sensitivity and specificity and the p-value probably to the AUROC in the Figure S4.

iv. S3+S5: Please clarify “Area Under the Curve” and “Coordinates of the Curve” below the description.

Our answer:

Thank you for this comment. We have removed Figures S2, S3, S5, and S6 from the Additional file. We have also deleted “Area Under the Curve” and “Coordinates of the Curve” below the description of Figure S5.

d. Figure S6:

i. Please introduce a space between “qSOFA” and “group” in the legend.

ii. Please consider avoiding the line description in the description, because of the provided legend.

iii. Please provide information on censoring, if present.

iv. Please provide the number of patients at risk for the qSOFA score groups below the time axis.

v. Please consider re-naming the time axis in “Days since …” with “...” naming the time origin.

vi. Please avoid abbreviations in the axes names.

vii. Please remove the title “Survival Function” above the plot.

Our answer:

Thank you for these comments. We have removed the Kaplan-Meier time-to-event analyses and the relevant results from the manuscript and supplementary file due to the mistakes mentioned in the comments above.

3) Tables:

a. Overall:

i. Please provide all abbreviations in the description.

Our answer:

Thank you for this comment. We have provided all abbreviations in the description.

ii. Please consider to write “p-value” instead of “p”.

Our answer:

Thank you for this comment. We have replaced “p” with “p-value”.

iii. Please use “(Q1, Q3)” instead of “(IQR)”, because the interquartile range is per definition the difference between Q3 and Q1, i.e. the third and the first quartile.

Our answer:

Thank you for this comment. We have replaced “(IQR)” with “(Q1-Q3)”.

iv. Please consider to provide consistently median with Q1 and Q3 instead of mean with SD for some characteristics.

Our answer:

Thank you for this comment. We have provided the consistently median with Q1 and Q3 instead of the mean with SD for some characteristics.

v. Please always provide the relative frequency with absolute counts, even in case of zero counts.

Our answer:

Thank you for this comment. We have provided the relative frequency with absolute counts, even in the case of zero counts.

vi. Please indicate whether a p-value for a categorical variable was derived from the chi2 or the exact Fisher test.

Our answer:

Thank you for this comment. We have indicated whether a p-value for a categorical variable was derived from the χ2 test or Fisher exact test.

b. Table S1:

i. Please consider “Documented comorbidities” instead of “Comorbidities”.

Our answer:

Thank you for this comment. We have replaced “Comorbidities” with “Documented comorbidities”.

ii. “SIRS” probably corresponds to the “SIRS criteria”. Please clarify.

Our answer:

Thank you for this comment. We have clarified the “SIRS criteria”

iii. Similarly, please consider adding the word “score” to “SOFA” and “qSOFA”.

Our answer:

Thank you for this comment. We have added the word “score” to “SOFA” and “qSOFA”.

iv. Regarding missing information/values, were all characteristics documented in all patients? If not, please provide the number of patients with missing information for a specific characteristic.

Our answer:

Thank you for this comment. Yes, they were. All characteristics were documented in all patients. However, we have also provided the number of patients with missing information for a specific variable in Table S9.

c. Table S2:

i. Line 3, first column: Probably “median (IQR) days” instead of “median (IQR), days” and “no. (%) of patients” instead of “no. (%)” to avoid misinterpretation.

Our answer:

Thank you for pointing this out. We have amended these typing mistakes.

ii. Please introduce the formats in this table. In case of missing information, the authors provide n/N (%), which should be stated somewhere. If the number of patients with information is constant across a section, the section heading might receive a footnote with the information on the number of patients with missing information.

Our answer:

Thank you for this comment. We have reformatted this table as required by the Reviewer.

d. Table S3:

i. See comments for Table S1.

Our answer:

Thank you for this comment. We have resolved the concerns raised by the Reviewer. 

ii. Please clarify “-“ in case of counts.

Our answer:

Thank you for this comment. Data were not available in these rows, so we have deleted them.

e. Table S4: See comments for Table S2.

Our answer:

Thank you for this comment. We have resolved the concerns raised by the Reviewer.

f. Table S5:

i. See comments for Table S1.

Our answer:

Thank you for this comment. We have resolved the concerns raised by the Reviewer.

ii. “bivariate regression analyses” indicates univariable logistic regression modelling? Please clarify.

Our answer:

Thank you for pointing this out. We had some typing mistakes in terms of logistic regression models. We performed the binary logistic regression analysis instead of the bivariate/univariable logistic regression analysis. We have clarified these issues throughout the manuscript.

iii. Please state the meaning of “-“.

Our answer:

Thank you for this comment. Data were not available in these rows, so we have deleted them.

iv. Please clarify “Frequency”. Please provide the number of patients included in the respective model.

Our answer:

Thank you for this comment. We have provided the number of patients included in the respective model.

v. Please clarify the difference between “>0.999” and “1.000” in the p-value column.

Our answer:

Thank you for this comment. The p-value is generally not found to be 1. It only becomes 1 when the values of both groups are the same (identical). However, when the p-value approaches near 1, e.g., 0.999, 0.998, or 0.989, it is regarded as 1. Our study presented some p-values at >0.999 because they are near 1.

g. Table S6:

i. Against the provided table, the authors included several independent variables in the model and performed a “multivariable logistic regression analyses” instead of a “multivariate logistic regression analyses” (several dependent variables). Please clarify.

Our answer:

Thank you for pointing this out. We had a typing mistake. We have clarified the variable selection.

ii. Please clarify “unit”. I would expect that individual level data was used and that for each variable the patient exhibited a specific category.

Our answer:

Thank you for this comment. In the multivariate logistic regression analysis, all potential candidate variables were re-coded to the categorical variables. Therefore, we report data on these categorical variables as percentages instead of units of measurement.

iii. Please mark the final model.

Our answer:

Thank you for this comment. We have marked the final model.

h. Table S7: See comments for Table S5.

Our answer:

Thank you for this comment. We have resolved the concerns raised by the Reviewer.

i. Table S8: See comments for Table S6.

Our answer:

Thank you for this comment. We have resolved the concerns raised by the Reviewer.

Reviewer #2

I would like to thank all authors for their wonderful work. It is very useful study. Mainly in developing countries managing sepsis.

Our answer:

Thank the Reviewer so much for taking the time to leave excellent reviews and positive feedback.

Minor comment:

1. In the discussion section please add few possible reasons, why SOFA score does not show higher prediction (Based on the ROC) in the selected sample.

Our answer:

Thank you for this comment. The present study shows that the predictive validity for hospital mortality of SOFA score was poor, with an AUROC of 0.688 (95% CI, 0.68 to 0.758), and is lower than those of the SOFA score (AUROC=0.74; 95% CI, 0.73 to 0.76) reported in the previously published study.[1] In this study, the SOFA score was calculated at the time of ICU admission. In contrast, the SOFA score in the previously published study was calculated for the time window from 48 hours before to 24 hours after the onset of an infection, as well as on each calendar day.[1] Therefore, these findings might explain why the predictive validity for hospital mortality of the SOFA score in our study was lower than those of the SOFA score reported in the previously published study. Because we have removed the AUROCs calculated to determine the discriminatory ability of the SOFA score for mortality away from the manuscript and Additional files, so we did not add this possible reason in the Discussion section.

[1] Seymour CW, Liu VX, Iwashyna TJ, Brunkhorst FM, Rea TD, Scherag A, Rubenfeld G, Kahn JM, Shankar-Hari M, Singer M, Deutschman CS, Escobar GJ, Angus DC. Assessment of Clinical Criteria for Sepsis: For the Third International Consensus Definitions for Sepsis and Septic Shock (Sepsis-3). JAMA. 2016 Feb 23;315(8):762-74. doi: 10.1001/jama.2016.0288. Erratum in: JAMA. 2016 May 24-31;315(20):2237. PMID: 26903335; PMCID: PMC5433435.

Reviewer #3

No suggestions. The manuscript is well written and can be accept in its present state. It is just advised the article is read and re-read to avoid any typos. A few typo errors were detected so a through proof reading is advised.

Other than that no suggestions are provided for improvement.

Our answer:

Thank the Reviewer so much for taking the time to leave excellent reviews and positive feedback.

Reviewer #4

Reviewer comments

Manuscript Number: PONE-D-21-40686

Title "Predictive validity of the quick Sequential Organ Failure Assessment (qSOFA) score for the mortality in patients with sepsis in Vietnamese intensive care units".

Generally speaking:

Thank you for providing me the opportunity to review this manuscript that raises important and interesting issues about predictive validity of the quick Sequential Organ Failure Assessment (qSOFA) score for the mortality in patients with sepsis in intensive care units in one of the developing countries.

Our answer:

Thank the Reviewer so much for taking the time to leave his/her excellent reviews.

Comment 1:

1. ABSTRACT:

a) No need for details of methodology in the abstract. Only main points. It should not include the type of statistical analysis.

Our answer:

Thank you for this comment. We have removed the type of statistical analysis (Lines 75-82).

b) Methodology has to mention the elements included as measuring parameter but without details.

Our answer:

Thank you for this comment. We have mentioned the primary and secondary outcomes in the Methods of the Abstract (Lines 77-78).

c) Abbreviations as APACHE, MOSAICS, and SIRS must be defined upon first appearance in the keywords.

Our answer:

Thank you for this comment. We have defined abbreviations (APACHE, MOSAICS, and SIRS) upon the first appearance in the keywords (Lines 95-97).

d) It is preferable that the keywords should be 5-6 words only.

Our answer:

Thank you for this comment. We have reduced the number of keywords to 6 words only (Lines 95-97).

Comment 2:

2. INTRODUCTION:

a) Global/ Regional/ Vietnam prevalence of the mortality of ICU patients with sepsis should be mentioned.

Our answer:

Thank you for this comment. In the Introduction section, we have mentioned the Global/ Regional/ Vietnam prevalence of the mortality of ICU patients with sepsis (Lines 110-116).

b) Factors associated with mortality in patients with sepsis upon Vietnamese intensive care unit admission should be clearly stated.

Our answer:

Thank you for this comment. We have clearly stated factors associated with mortality in patients with sepsis upon Vietnamese intensive care unit admission (Lines 139-144).

c) Explaining why this topic was chosen for analysis in this article is not well written. The benefits of conducting the study to the community should be explained.

Our answer:

Thank you for pointing this out. We have rewritten the benefits of conducting the study for the community and clearly explained why the topic was chosen for analysis in this article (Lines 145-147).

Comment 3:

3. METHODS:

a) Were the samples normal or not?

Our answer:

Thank you for this valuable comment. Yes, the distribution of the sample was normal. In this cross-sectional study, the primary outcome was hospital mortality. Therefore, based on the hospital mortality rate (61.0%) of our cohort reported in a previously published study,[1] we used the formula to find the minimum sample size for estimating a population proportion, with a confidence level of 95% and a confidence interval (margin of error) of ±6.03%, and an assumed population proportion of 61.0%. As a result, our sample size should be at least 252 patients. Therefore, our sample was large enough size which reflects a normal distribution. We have provided the sample size calculation in the Methods section (Lines 199-210). However, although the sample size was large enough, the confidence interval was slightly wide (±6.03%), which might influence the normal distribution of the sample. Therefore, we have added this limitation to the Strengths and limitations section (Lines 381-383).

[1] Thao PTN, Tra TT, Son NT, Wada K. Reduction in the IL-6 level at 24 h after admission to the intensive care unit is a survival predictor for Vietnamese patients with sepsis and septic shock: a prospective study. BMC Emerg Med. 2018 Nov 6;18(1):39. doi: 10.1186/s12873-018-0191-4. PMID: 30400775; PMCID: PMC6219151.

b) Tests of significant for each type of variables should be mentioned.

Our answer:

Thank you for this comment. We have mentioned the tests of significance for each type of variables (Lines 212-218).

c) In lines 158 and 159, why predominantly neurosurgical, coronary, and cardiothoracic ICUs were excluding?

Our answer:

Thank you for this comment. In this study, we defined an intensive care unit (ICU) as an ICU capable of providing invasive mechanical ventilation and organ support, such as using vasoactive medications and renal replacement therapy, and recognized to be an ICU by its hospital. However, the predominantly neurosurgical, coronary, and cardiothoracic ICUs often include neurosurgical or cardiac patients. Moreover, the present study defined a neurosurgical patient as a patient admitted to the neurosurgical ICU after or pending neurosurgical interventions and defined a cardiac patient as a patient admitted to the coronary and cardiothoracic ICU after cardiac surgery for acute myocardial infarction without shock or respiratory failure, and for arrhythmias. Therefore, to limit biases (e.g., selection bias), we excluded predominantly neurosurgical, coronary, and cardiothoracic ICUs.

d) The basis of sample size calculation should be clearly stated to know the confidence level and the margin of error.

Our answer:

Thank you for this comment. We have provided the sample size calculation in the Methods section (Lines 199-210).

Comment 4:

4. RESULTS:

a) No need to conduct bivariate logistic regression because it doesn’t consider the confounding variables.

Our answer:

Thank you for this comment.

b) What were the criteria to pass your variables from bivariate to multivariate logistic regression?

Our answer:

Thank you for this comment. We have described more clearly the variable selection and answered the Reviewer's questions (Lines 224-241).

Comment 5:

5. DISCUSSION:

a) It is advisable to explain the study objective at the beginning of the discussion.

Our answer:

Thank you for this comment. We have explained the study objective at the beginning of the discussion (Lines 309-312).

b) Compare the findings of the study with other findings and state the reasons for the strengths and weaknesses in each section. The manuscript could be greatly strengthened if the authors could provide highlight on the main and significant factors associated with hospital and ICU mortalities in patients with sepsis in other developing and developed countries with similar context.

Our answer:

Thank you for this valuable comment. We have compared the findings between the present study and other studies and stated the reasons for the strengths and weaknesses in each section. We also have highlighted the main and significant factors associated with the hospital and ICU mortalities in patients with sepsis in other developing and developed countries with similar contexts (Lines 313-339).

c) The manuscript could also be greatly strengthened if the authors could provide highlight on the main and significant life-sustaining treatments during ICU stay and outcomes of ICU patients with sepsis according to hospital survivability in other developing and developed countries with similar context.

Our answer:

Thank you for this valuable comment. We have highlighted the main and significant life-sustaining treatments during ICU stay and outcomes of ICU patients with sepsis according to hospital survivability in other developing and developed countries with similar contexts (Lines 326-331).

d) In brief, discuss by using the scientific reasoning the differences in accuracy of the qSOFA score to predict the risk of hospital and ICU deaths from sepsis.

Our answer:

Thank you for this comment.

e) In lines 305 and 306, the sentence “Our 304 figures for the hospital and ICU mortality rates are in line with the figures reported in the 305 worldwide Intensive Care over Nations (ICON) study (19.3% [26/439] to 47.2% [17/141], and 11.9% [16/439] to 39.5 [15/141], respectively)” was not clearly stated.

Our answer:

Thank you for this comment. We have removed this statement.

f) Abbreviations as HIC must be defined upon first appearance in the discussion.

Our answer:

Thank you for this comment. We have defined HIC as an abbreviation upon the first appearance in the discussion (Lines 322-323).

Comment 6:

6. CONCLUSION:

Please write suggestions for improvement.

Our answer:

Thank you for this comment. We have written suggestions for improvement (Lines 389-391).

Comment 7:

7. STRENGTHS AND LIMITATIONS:

Strengths and limitations of the study should be analyzed in separate paragraphs.

Our answer:

Thank you for this comment. We have analyzed the strengths and limitations of the study in separate paragraphs (Lines 370-383).

Thank the Editors and Reviewers so much for taking the time to leave their excellent reviews!

Sincerely yours,

Chinh Quoc Luong, MD., PhD.

Center for Emergency Medicine,

Bach Mai Hospital,

No. 78, Giai Phong, Phuong Mai ward, Dong Da district, Hanoi 100000, Vietnam

Email: luongquocchinh@gmail.com

---

## [Decision Letter · Decision Letter 1]

16 Aug 2022

PONE-D-21-40686R1Predictive validity of the quick Sequential Organ Failure Assessment (qSOFA) score for the mortality in patients with sepsis in Vietnamese intensive care unitsPLOS ONE

Dear Dr. Luong,

Thank you for submitting your manuscript to PLOS ONE. After careful consideration, we feel that it has merit but does not fully meet PLOS ONE’s publication criteria as it currently stands. Therefore, we invite you to submit a revised version of the manuscript that addresses the points raised during the review process.

We look forward to receiving your revised manuscript.

Kind regards,

Dinh-Toi Chu, PhD

Academic Editor

PLOS ONE

Journal Requirements:

Reviewers' comments:

Reviewer's Responses to Questions

**Comments to the Author**

1. If the authors have adequately addressed your comments raised in a previous round of review and you feel that this manuscript is now acceptable for publication, you may indicate that here to bypass the “Comments to the Author” section, enter your conflict of interest statement in the “Confidential to Editor” section, and submit your "Accept" recommendation.

Reviewer #1: (No Response)

Reviewer #2: All comments have been addressed

Reviewer #4: All comments have been addressed

2. Is the manuscript technically sound, and do the data support the conclusions?

Reviewer #1: Partly

Reviewer #2: Yes

Reviewer #4: Yes

3. Has the statistical analysis been performed appropriately and rigorously? 

Reviewer #1: I Don't Know

Reviewer #2: Yes

Reviewer #4: Yes

4. Have the authors made all data underlying the findings in their manuscript fully available?

Reviewer #1: No

Reviewer #2: Yes

Reviewer #4: Yes

5. Is the manuscript presented in an intelligible fashion and written in standard English?

Reviewer #1: Yes

Reviewer #2: Yes

Reviewer #4: Yes

6. Review Comments to the Author

Reviewer #1: The authors have greatly improved their manuscript. However, some issues remain to be addressed.

Major issues:

1) The authors name their analyses “multivariate”, but I still do not identify the multivariate aspects. The authors analysed ICU and hospital mortality separately. Hence, they consider only one dependent variable at a time. Regarding the independent variables in the regression models, the authors build univariable and multivariable regression models, i.e. including one (univariable) or more than one (multivariable) independent variable. So, please clarify.

2) One of my previous comments might have been not clear enough. As I asked about the way, according to which possible centre effects were accounted for in the regression models, this was not about the variable central hospital (yes/no), but about mixed effect logistic regression modelling with the centre as random effect. Did the authors consider this approach? If so, why did the authors decided not to apply it?

3) The description of the modelling approach is still not completely clear.

a. Please clarify “candidate variables” in line 241/242.

b. Please clarify “some significant contribution to the outcome”. Which criteria apply? Please add all required information in the method and result section.

c. Is there a link between step (b) and (c)? If so, please state. If not, some results are probably missing. Please clarify.

4) It is not clear, whether the authors additionally applied the originally proposed cut-off value for the qSOFA score to their own data. If this was done, please add this in the method and the results section. Otherwise, please perform this additional analysis and report it in the manuscript. Subsequently, this can be added to the discussion as well.

Minor issues:

1) Line 82: Please add “in univariable and multivariable regression modelling” behind “factors associated with the hospital and ICU mortalities were assessed”. Otherwise, the reader is surprised by the results (AOR) in the subsequent section.

2) Line 76/77: Please consider rewording. Suggestion: “[…] Vietnam on specified days […] representing the four different seasons of 2019.”

3) Line 90: OR should probably be AOR.

4) Line 128: “normal ward” instead of “ward”, if this applies.

5) Line 156: One full stop too much. Please remove.

6) Line 160: Please clarify “representatives”. It becomes not clear, whether they are part of the local study team.

7) Line 162: Please rephrase. Suggestion: “[…] on one of the four days […]”.

8) Line 163: Please add 2019 behind the dates.

9) Line 171: Please clarify. Suggestion: “Data was entered by the representatives of the hospitals […]”, if this applies. Otherwise, please clarify.

10) Line 192: “or” instead of “and”?

11) Line 216: Please write chi-2 with a 2 as superscript.

12) Please unify the number of decimal places. The number should be consistent for all median values and corresponding quartiles, for all percentages (always one decimal place, even for 0.0%) and so on. This applies for the text body as well as for tables and figures.

13) Please consider to write p_AUROC (_ should indicate, that AUROC should be a subscript) instead of p for p-values corresponding to AUCs. This would facilitate reading. This applies to main text as well as figure and table descriptions.

14) Table 1:

a. Please indicate, if for a variable no p-value is provided (here: HIV infection), e.g. by -.

b. Abbreviations n and no. are missing. Please add.

15) Table 2: Abbreviations n and no. are missing. Please add.

16) Table 3:

a. “Binary” is probably “univariable” and “multivariate” is probably “multivariable”. See comment above. Please adapt, if this applies. Otherwise, please clarify.

b. Which kind of multivariable regression modelling was applied? Results of final model from backward elimination?

c. Please provide the reference to the complete regression results in the description. At least several univariable models are probably missing. If results from backward elimination are presented, then here should also be a link to the respective results in the supplement.

17) Table S1:

a. There seems to be some redundancy with Table 1. Please clarify. Is Table 1 just the short version for the main text body? Then, please refer in the description of Table 1 for additional information to Table S1.

b. Abbreviations n and no. are missing. Please add.

18) Table S2:

a. There seems to be some redundancy with Table 2. Please check/clarify. Is Table 2 just the short version for the main text body? Then, please refer in in the description of Table 2 for additional information to Table S2.

b. “no. (%) and median (Q1-Q3), days”:

i. “and” should be “or”. Please check.

ii. The comma should be removed. Please check.

c. Abbreviations n and no. are missing. Please add.

19) Table S3: Abbreviations n and no. are missing. Please add.

20) Table S4:

a. “no. (%) and median (Q1-Q3) days”: “and” should be “or”.

b. Abbreviations n and no. are missing. Please add.

21) Table S5: Please see for “bivariate regression analyses” comments above.

22) Table S6: Please see for “bivariate regression analyses” comments above.

23) Table S7:

a. Please see for “multivariate logistic regression analyses” comments above.

b. Please add the number of patients included in the models.

c. “Unit” is still not clear. Was modelling performed on individual patients’ data or already on aggregated data? If the analysis was done on individual level data, it is unclear, how a patient, for example, can only be in parts older than 65 years of age. A patient is either below 65 years or above, but cannot be a mixture of both. These questions are also related to the comment above on a more detailed method description.

24) Table S8: Please see comments for Table S7.

25) Figure 1: Please introduce the word “abbreviations:” in front of the list of abbreviation (as in Figure S1).

26) Figure S2: Please see comment for Figure 1.

27) Figure S3: Please see comment for Figure 1.

28) Figure S4: Please see comment for Figure 1.

Reviewer #2: (No Response)

Reviewer #4: Reviewer comments

Manuscript Number: PONE-D-21-40686_R1

Title "Predictive validity of the quick Sequential Organ Failure Assessment (qSOFA) score for the mortality in patients with sepsis in Vietnamese intensive care units".

Thank you for providing me the opportunity to review this revised manuscript that raises important issues about predictive validity of the quick Sequential Organ Failure Assessment (qSOFA) score for the mortality in patients with sepsis in intensive care units in one of the developing countries.

It seems that all corrections were done.

7. PLOS authors have the option to publish the peer review history of their article (what does this mean?). If published, this will include your full peer review and any attached files.

Reviewer #1: **Yes: **Miriam Kesselmeier

Reviewer #2: No

Reviewer #4: No

---

## [Author Response · Author response to Decision Letter 1]

1 Sep 2022

Professor Dinh-Toi Chu, PhD.

Academic Editor

PLOS ONE

August 21, 2022

Dear Prof. Dinh-Toi Chu,

On behalf of all authors, I am resubmitting herewith our revised manuscript entitled “Predictive validity of the quick Sequential Organ Failure Assessment (qSOFA) score for the mortality in patients with sepsis in Vietnamese intensive care units” (PONE-D-21-40686R1).

We sincerely appreciate the kind comments and points raised by the Editors and the Reviewers. We have carefully considered all comments and suggestions and revised our manuscript following these points. These comments have greatly enabled us to improve our manuscript. We hope that Editor will find our revised manuscript suitable for publication in PLOS ONE.

We confirm that this work is original and has not been published elsewhere, nor is it currently under consideration for publication elsewhere. All authors have read, approved the revised manuscript, and agreed to authorship and order of authorship for this manuscript. All authors have the appropriate permissions and rights to the reported data.

We have provided our point-by-point responses to the comments of the Editors and the Reviewers below.

We thank you for your kind consideration of this submission.

Sincerely yours,

Chinh Quoc Luong, MD., PhD.

Center for Emergency Medicine,

Bach Mai Hospital,

No. 78, Giai Phong, Phuong Mai ward, Dong Da district, Hanoi 100000, Vietnam

Email: luongquocchinh@gmail.com

We thank the Editors and the Reviewers for the valuable comments and suggestions that greatly helped us to improve the contents of this paper. In what follows, we will use the boldface to indicate comments from the Editors and the Reviewers, the standard font face for our responses and we highlighted in yellow the modifications that we did to the manuscript.

RESPONSE TO EDITORS

Dear Dr. Luong,

Thank you for submitting your manuscript to PLOS ONE. After careful consideration, we feel that it has merit but does not fully meet PLOS ONE’s publication criteria as it currently stands. Therefore, we invite you to submit a revised version of the manuscript that addresses the points raised during the review process.

Our answer:

We thank you for the positive feedback. We have carefully considered the Reviewers' comments and suggestions and have revised our manuscript following each of these points.

Thank you for this comment. We have submitted the revised manuscript on time.

Our answer:

Thank you for this comment. We have included a rebuttal letter, a marked-up copy of the manuscript, and an unmarked version of the revised manuscript when submitting our revised manuscript.

Our answer:

Thank you for this comment. We do not have any changes in our financial disclosure.

Our answer:

Thank you for this comment. Laboratory protocol does not apply to our study, but study protocol does. Our study protocol has included in the Methods section.

Journal Requirements:

Our answer:

Thank you for this comment. We have reviewed the reference list, which is complete and correct. There are not any retracted articles on the list.

RESPONSE TO REVIEWERS

6. Review Comments to the Author

Reviewer #1: The authors have greatly improved their manuscript. However, some issues remain to be addressed.

Our answer:

Thank the Reviewer so much for taking the time to leave the excellent reviews!

Major issues:

1) The authors name their analyses “multivariate”, but I still do not identify the multivariate aspects. The authors analysed ICU and hospital mortality separately. Hence, they consider only one dependent variable at a time. Regarding the independent variables in the regression models, the authors build univariable and multivariable regression models, i.e. including one (univariable) or more than one (multivariable) independent variable. So, please clarify.

Our answer:

Thank you for this valuable comment that helped us understands further the terms of regression models. In each of our regression models, there was only one dependent variable (ICU or hospital mortality) at a time; thus, we have replaced the terms 'binary/univariate' or 'multivariate' with 'univariable' or 'multivariable' for these models throughout the manuscript.

2) One of my previous comments might have been not clear enough. As I asked about the way, according to which possible centre effects were accounted for in the regression models, this was not about the variable central hospital (yes/no), but about mixed effect logistic regression modelling with the centre as random effect. Did the authors consider this approach? If so, why did the authors decided not to apply it?

Our answer:

We are sorry for misunderstanding your comment in the previous revision of the manuscript. The concern of possible center effects, which you have raised, is valuable for our paper. To the best of our knowledge, the mixed-effects logistic regression model can be used to predict discrete outcome variables when observations are correlated. In our study, hospital mortality (dependent variable) was a primary outcome measured at two different time points, i.e., inside and outside the intensive care unit (ICU). Therefore, mixed effect logistic regression modeling with the center as a random effect is the most suitable for our analysis. However, our independent variables (e.g., qSOFA) were not measured at two corresponding time points. Therefore, this limitation prevented us from applying a mixed-effects logistic regression model. We have added this issue to the Limitation section as follows:

“Due to our independent variables (e.g., qSOFA) that might be associated with primary outcome (hospital mortality) only measured upon ICU admission, the mixed-effects logistic regression model could not be used to predict discrete outcome variables measured at two different times, i.e., inside and outside the ICU.” (Lines 403-407)

3) The description of the modelling approach is still not completely clear.

a. Please clarify “candidate variables” in line 241/242.

Our answer:

Thank you for this comment. We have clarified “candidate variables” and have added this sentence to the Statistical analyses section as follows:

“Using a stepwise backward elimination method, we started with the full multivariable logistic regression model that included the selected variables.” (Lines 245-247)

b. Please clarify “some significant contribution to the outcome”. Which criteria apply? Please add all required information in the method and result section.

Our answer:

Thank you for this comment. We have clarified “some significant contribution to the outcome” and have added this sentence to the Statistical analyses section as follows:

“This method then deleted variables stepwise from the full model until all remaining variables were independently associated with the risk of death in the hospital.” (Lines 247-249)

c. Is there a link between step (b) and (c)? If so, please state. If not, some results are probably missing. Please clarify.

Our answer:

Thank you for this comment. There was a link between steps (b) and (c). Therefore, we have merged these steps into one and have stated them in the Statistical analyses section (Lines 239-249).

4) It is not clear, whether the authors additionally applied the originally proposed cut-off value for the qSOFA score to their own data. If this was done, please add this in the method and the results section. Otherwise, please perform this additional analysis and report it in the manuscript. Subsequently, this can be added to the discussion as well.

Our answer:

Thank you for this valuable comment that helped us improve the contents of this paper. We have performed additional analysis and reported it in the Method section (Lines 232-234), the Results section (Lines 311-316), and the Discussion section (Lines 382-388) of the manuscript.

Minor issues:

1) Line 82: Please add “in univariable and multivariable regression modelling” behind “factors associated with the hospital and ICU mortalities were assessed”. Otherwise, the reader is surprised by the results (AOR) in the subsequent section.

Our answer:

Thank you for this comment. We have added “in univariable and multivariable logistic models” behind “factors associated with the hospital and ICU mortalities were assessed” (Lines 82-83).

2) Line 76/77: Please consider rewording. Suggestion: “[…] Vietnam on specified days […] representing the four different seasons of 2019.”

Our answer:

Thank you for this comment. We have reworded this sentence (Line 76).

3) Line 90: OR should probably be AOR.

Our answer:

Thank you for this comment. This error is typing mistake. We have replaced ‘OR’ with ‘AOR’ (Line 91).

4) Line 128: “normal ward” instead of “ward”, if this applies.

Our answer:

Thank you for this comment. We have replaced ‘ward’ with ‘normal ward’ (Line 129).

5) Line 156: One full stop too much. Please remove.

Our answer:

Thank you for this comment. We have removed a redundant punctuation mark (Line 157).

6) Line 160: Please clarify “representatives”. It becomes not clear, whether they are part of the local study team.

Our answer:

Thank you for this comment. Representatives are part of the local study team and the MOSAICS II study group. We have clarified ‘representatives’ as follows:

“Each ICU had one or two representatives who were part of the local study team and the MOSAICS II study group, as shown in eAppendix 2 of a previously published paper.[6]” (Lines 160-162)

7) Line 162: Please rephrase. Suggestion: “[…] on one of the four days […]”.

Our answer:

Thank you for this comment. We have rephrased this sentence. (Line 165)

8) Line 163: Please add 2019 behind the dates.

Our answer:

Thank you for this comment. We have added ‘2019’ behind the dates. (Line 166)

9) Line 171: Please clarify. Suggestion: “Data was entered by the representatives of the hospitals […]”, if this applies. Otherwise, please clarify.

Our answer:

Thank you for this comment. We have clarified this sentence. (Lines 175-176)

10) Line 192: “or” instead of “and”?

Our answer:

Thank you for this comment. We have replaced 'and' with 'or'. (Line 197)

11) Line 216: Please write chi-2 with a 2 as superscript.

Our answer:

Thank you for this comment. We have rewritten this word. (Lines 220-221)

12) Please unify the number of decimal places. The number should be consistent for all median values and corresponding quartiles, for all percentages (always one decimal place, even for 0.0%) and so on. This applies for the text body as well as for tables and figures.

Our answer:

Thank you for pointing this out. We have unified the number of decimal places throughout the manuscript as well as the tables and figures; the number has been consistent for all median values and corresponding quartiles, for all percentages, and so on.

13) Please consider to write p_AUROC (_ should indicate, that AUROC should be a subscript) instead of p for p-values corresponding to AUCs. This would facilitate reading. This applies to main text as well as figure and table descriptions.

Our answer:

Thank you for this comment. We have written PAUROC instead of p for p-values corresponding to AUCs throughout the manuscript, as well as figure and table descriptions.

14) Table 1:

a. Please indicate, if for a variable no p-value is provided (here: HIV infection), e.g. by -.

Our answer:

Thank you for this comment. We have added an abbreviation ‘NA’ that means not available for where no p-values provided.

b. Abbreviations n and no. are missing. Please add.

Our answer:

Thank you for this comment. We have added abbreviations ‘n’ and ‘no.’

15) Table 2: Abbreviations n and no. are missing. Please add.

Our answer:

Thank you for this comment. We have added abbreviations ‘n’ and ‘no.’

16) Table 3:

a. “Binary” is probably “univariable” and “multivariate” is probably “multivariable”. See comment above. Please adapt, if this applies. Otherwise, please clarify.

Our answer:

Thank you for this valuable comment that helped us understands further the terms of regression models. In each of our regression models, there was only one dependent variable (ICU or hospital mortality) at a time; thus, we have replaced the terms ‘binary/univariate’ or ‘multivariate’ with ‘univariable’ or ‘multivariable’ for these models throughout the manuscript, as well as table descriptions.

b. Which kind of multivariable regression modelling was applied? Results of final model from backward elimination?

Our answer:

Thank you for this comment. We used the multivariable logistic regression model with the stepwise backward elimination method. Table 3 presents the results of the final multivariable logistic regression model from backward elimination. We have clarified this issue at the bottom of the table.

c. Please provide the reference to the complete regression results in the description. At least several univariable models are probably missing. If results from backward elimination are presented, then here should also be a link to the respective results in the supplement.

Our answer:

Thank you for this comment. We have provided the reference to the complete regression results in the description.

17) Table S1:

a. There seems to be some redundancy with Table 1. Please clarify. Is Table 1 just the short version for the main text body? Then, please refer in the description of Table 1 for additional information to Table S1.

Our answer:

Thank you for this comment. Yes, it is. Table 1 is just the short version of the main text body. We have referred to the description of Table 1 for additional information to Table S1.

b. Abbreviations n and no. are missing. Please add.

Our answer:

Thank you for this comment. We have added abbreviations ‘n’ and ‘no.’

18) Table S2:

a. There seems to be some redundancy with Table 2. Please check/clarify. Is Table 2 just the short version for the main text body? Then, please refer in in the description of Table 2 for additional information to Table S2.

Our answer:

Thank you for this comment. Yes, it is. Table 2 is just the short version of the main text body. We have referred to the description of Table 2 for additional information to Table S2.

b. “no. (%) and median (Q1-Q3), days”:

i. “and” should be “or”. Please check.

Our answer:

Thank you for this comment. We have replaced 'and' with 'or'.

ii. The comma should be removed. Please check.

Our answer:

Thank you for this comment. We have removed the comma.

c. Abbreviations n and no. are missing. Please add.

Our answer:

Thank you for this comment. We have added abbreviations ‘n’ and ‘no.’

19) Table S3: Abbreviations n and no. are missing. Please add.

Our answer:

Thank you for this comment. We have added abbreviations ‘n’ and ‘no.’

20) Table S4:

a. “no. (%) and median (Q1-Q3) days”: “and” should be “or”.

Our answer:

Thank you for this comment. We have replaced 'and' with 'or'.

b. Abbreviations n and no. are missing. Please add.

Our answer:

Thank you for this comment. We have added abbreviations ‘n’ and ‘no.’

21) Table S5: Please see for “bivariate regression analyses” comments above.

Our answer:

Thank you for this valuable comment that helped us understands further the terms of regression models. In each of our regression models, there was only one dependent variable (ICU or hospital mortality) at a time; thus, we have replaced the terms ‘binary/univariate’ or ‘multivariate’ with ‘univariable’ or ‘multivariable’ for these models throughout the manuscript, as well as table descriptions.

22) Table S6: Please see for “bivariate regression analyses” comments above.

Our answer:

Thank you for this valuable comment that helped us understands further the terms of regression models. In each of our regression models, there was only one dependent variable (ICU or hospital mortality) at a time; thus, we have replaced the terms ‘binary/univariate’ or ‘multivariate’ with ‘univariable’ or ‘multivariable’ for these models throughout the manuscript, as well as table descriptions.

23) Table S7:

a. Please see for “multivariate logistic regression analyses” comments above.

Our answer:

Thank you for this valuable comment that helped us understands further the terms of regression models. In each of our regression models, there was only one dependent variable (ICU or hospital mortality) at a time; thus, we have replaced the terms ‘binary/univariate’ or ‘multivariate’ with ‘univariable’ or ‘multivariable’ for these models throughout the manuscript, as well as table descriptions.

b. Please add the number of patients included in the models.

Our answer:

Thank you for this comment. We have added the number of patients included in the models.

c. “Unit” is still not clear. Was modelling performed on individual patients’ data or already on aggregated data? If the analysis was done on individual level data, it is unclear, how a patient, for example, can only be in parts older than 65 years of age. A patient is either below 65 years or above, but cannot be a mixture of both. These questions are also related to the comment above on a more detailed method description.

Our answer:

Thank you for this comment. We have replaced 'Unit' with 'Number of patients'.

24) Table S8: Please see comments for Table S7.

Our answer:

Thank you for this comment. We have addressed the concerns you raised.

25) Figure 1: Please introduce the word “abbreviations:” in front of the list of abbreviation (as in Figure S1).

Our answer:

Thank you for this comment. We have addressed the concerns you raised.

26) Figure S2: Please see comment for Figure 1.

Our answer:

Thank you for this comment. We have addressed the concerns you raised.

27) Figure S3: Please see comment for Figure 1.

Our answer:

Thank you for this comment. We have addressed the concerns you raised.

28) Figure S4: Please see comment for Figure 1.

Our answer:

Thank you for this comment. We have addressed the concerns you raised.

Reviewer #2: (No Response)

We thank the Reviewer for the valuable comments and suggestions that helped us improve the contents of this paper.

Reviewer #4: Reviewer comments

Manuscript Number: PONE-D-21-40686_R1

Title "Predictive validity of the quick Sequential Organ Failure Assessment (qSOFA) score for the mortality in patients with sepsis in Vietnamese intensive care units".

Thank you for providing me the opportunity to review this revised manuscript that raises important issues about predictive validity of the quick Sequential Organ Failure Assessment (qSOFA) score for the mortality in patients with sepsis in intensive care units in one of the developing countries.

It seems that all corrections were done.

Our answer:

We thank the Reviewer for the valuable comments and suggestions that helped us improve the contents of this paper.

Thank the Editors and Reviewers so much for taking the time to leave their excellent reviews!

Sincerely yours,

Chinh Quoc Luong, MD., PhD.

Center for Emergency Medicine,

Bach Mai Hospital,

No. 78, Giai Phong, Phuong Mai ward, Dong Da district, Hanoi 100000, Vietnam

Email: luongquocchinh@gmail.com

---

## [Editor Report · Decision Letter 2]

22 Sep 2022

Predictive validity of the quick Sequential Organ Failure Assessment (qSOFA) score for the mortality in patients with sepsis in Vietnamese intensive care units

PONE-D-21-40686R2

Dear Dr. Luong,

We’re pleased to inform you that your manuscript has been judged scientifically suitable for publication and will be formally accepted for publication once it meets all outstanding technical requirements.

Kind regards,

Dinh-Toi Chu, PhD

Academic Editor

PLOS ONE
---

## [Editor Report · Acceptance letter]

3 Oct 2022

PONE-D-21-40686R2 

Predictive validity of the quick Sequential Organ Failure Assessment (qSOFA) score for the mortality in patients with sepsis in Vietnamese intensive care units 

Dear Dr. Luong:

I'm pleased to inform you that your manuscript has been deemed suitable for publication in PLOS ONE. Congratulations! Your manuscript is now with our production department. 

Kind regards, 

on behalf of

Dr. Dinh-Toi Chu 

Academic Editor

PLOS ONE